# Spatial organization of adenylyl cyclase and its impact on dopamine signaling in neurons

Léa Ripoll [1,2], Yong Li[3], Carmen W. Dessauer [3] & Mark von Zastrow [1,2,4] ✉

The cAMP cascade is increasingly recognized to transduce physiological effects locally through spatially limited cAMP gradients. However, little is known about how adenylyl cyclase enzymes that initiate cAMP gradients are localized. Here we address this question in physiologically relevant striatal neurons and investigate how AC localization impacts downstream signaling function. We show that the major striatal AC isoforms are differentially sorted between ciliary and extraciliary domains of the plasma membrane, and that one isoform, AC9, is uniquely concentrated in endosomes. We identify key sorting determinants in the N-terminal cytoplasmic domain responsible for isoform-specific localization. We further show that AC9-containing endosomes accumulate activated dopamine receptors and form an elaborately intertwined network with juxtanuclear PKA stores bound to Golgi membranes. Finally, we provide evidence that endosomal localization enables AC9 to selectively elevate PKA activity in the nucleus relative to the cytoplasm. Together, these results reveal a precise spatial landscape of the cAMP cascade in neurons and a key role of AC localization in directing downstream PKA signaling to the nucleus.

Cyclic AMP (cAMP) is a pivotal second messenger that transduces extracellular chemical cues recognized by G protein-coupled receptors (GPCRs) on the cell surface into downstream physiological responses inside the cell. Despite its discovery as a diffusible second messenger, many physiological effects of cAMP are now thought to occur through the formation of spatially restricted cAMP gradients[1,2]. It is increasingly evident that local cAMP signaling is not limited to the plasma membrane, as GPCRs can also be activated from different subcellular locations and produce distinct downstream cellular responses[3–7]. To date, efforts to understand local cAMP compartmentalization have focused primarily on phosphodiesterases (PDEs) that hydrolyze cAMP and on A-kinase anchoring proteins (AKAPs) that sequester cAMP-dependent protein kinase (PKA) and downstream signaling enzymes in nanodomains[7–9]. However, a key prerequisite for local signaling to occur is the local production of cAMP, yet relatively little is known about how the adenylyl cyclase (AC) enzymes that produce cAMP are localized within cells.

Nine homologous AC isoforms (AC1–AC9) mediate GPCR-regulated cAMP production in mammals. All are integral membrane proteins whose enzymatic activity is stimulated by the heterotrimeric G proteins $G\alpha_{s/olf}$, but each differs in additional regulation by other biochemical inputs such as $G\alpha_{i/o}$ proteins and calcium[10]. Much of what is presently known about AC localization concerns AC partitioning at the plasma membrane between lateral domains (e.g., 'rafts' vs. 'non-rafts')[11,12] or scaffolding by AKAPs into signaling complexes[13,14]. Accumulating evidence indicates that AC isoforms are also functionally organized at internal membrane locations, but how such isoform-specific localization at distinct membranes is achieved in physiologically relevant cells remains largely unknown.

Evidence for local cAMP signaling is particularly strong in neurons[15–21]. In the striatum, for example, dopamine regulates motor and motivated behaviors through cAMP-mediated control of PKA activity in medium spiny projection neurons (MSNs). Interestingly, disrupting PKA compartmentalization in striatal MSNs, by knocking-

[1]Department of Psychiatry and Behavioral Sciences, University of California, San Francisco, San Francisco, CA, USA. [2]Department of Cellular and Molecular Pharmacology, University of California, San Francisco, San Francisco, CA, USA. [3]Department of Integrative Biology and Pharmacology, McGovern Medical School, University of Texas Health Science Center, Houston, TX, USA. [4]Quantitative Biology Institute, University of California, San Francisco, San Francisco, CA, USA. ✉e-mail: mark@vzlab.org

out PKA RIIβ that localizes PKA through binding to AKAPs, disrupts long-term dopaminergic control of cAMP-dependent gene expression without impairing short-term motor control[22]. This suggests that localized cAMP-PKA signaling contributes to determining the physiological effects of dopamine on neural plasticity[23].

Neurons typically co-express multiple AC isoforms, each with distinct regulatory properties[24,25]. In the striatum, AC5 is responsible for the majority of the cAMP production[26], but AC3 and AC9 are also expressed[25,27,28]. Interestingly, AC5 knock-out in mouse striatum blocks some but not all physiological actions of dopamine[29], and it selectively impairs some but not all forms of striatum-dependent learning[30]. Together, these observations raise two fundamental questions: First, are distinct AC isoforms that are natively co-expressed in MSNs differentially localized at the subcellular level? Second, if so, how does subcellular location impact downstream signaling?

Here, we delineate a precise spatial landscape of isoform-specific localization in MSNs and provide mechanistic insight to how these differences are determined. Focusing on AC9, which is uniquely targeted to endosomes, we identify a functional impact of specific localization by revealing an essential role of AC9 targeting to endosomes in mediating downstream PKA signaling to the nucleus.

## Results

### Isoform-specific targeting of striatal ACs in the plasma membrane and endosomes

To begin investigating the subcellular distribution of the selected striatal AC isoforms, we constructed N-terminally HA-tagged versions of each isoform and visualized their subcellular localization in cultured striatal MSNs. Confocal imaging of fixed cells localized HA-AC3 predominantly on the neuronal cell body in a hair-like microdomain that we identified as the primary cilium by colocalization with the ciliary membrane marker Arl13b (Fig. 1a), consistent with previous reports[28,31]. HA-AC5 localized in the ciliary microdomain on the soma but was also present in the extraciliary plasma membrane, both on the soma and processes (Fig. 1b). HA-AC9, in contrast, was distributed on the neuron surface exclusively outside of the ciliary microdomain and localized

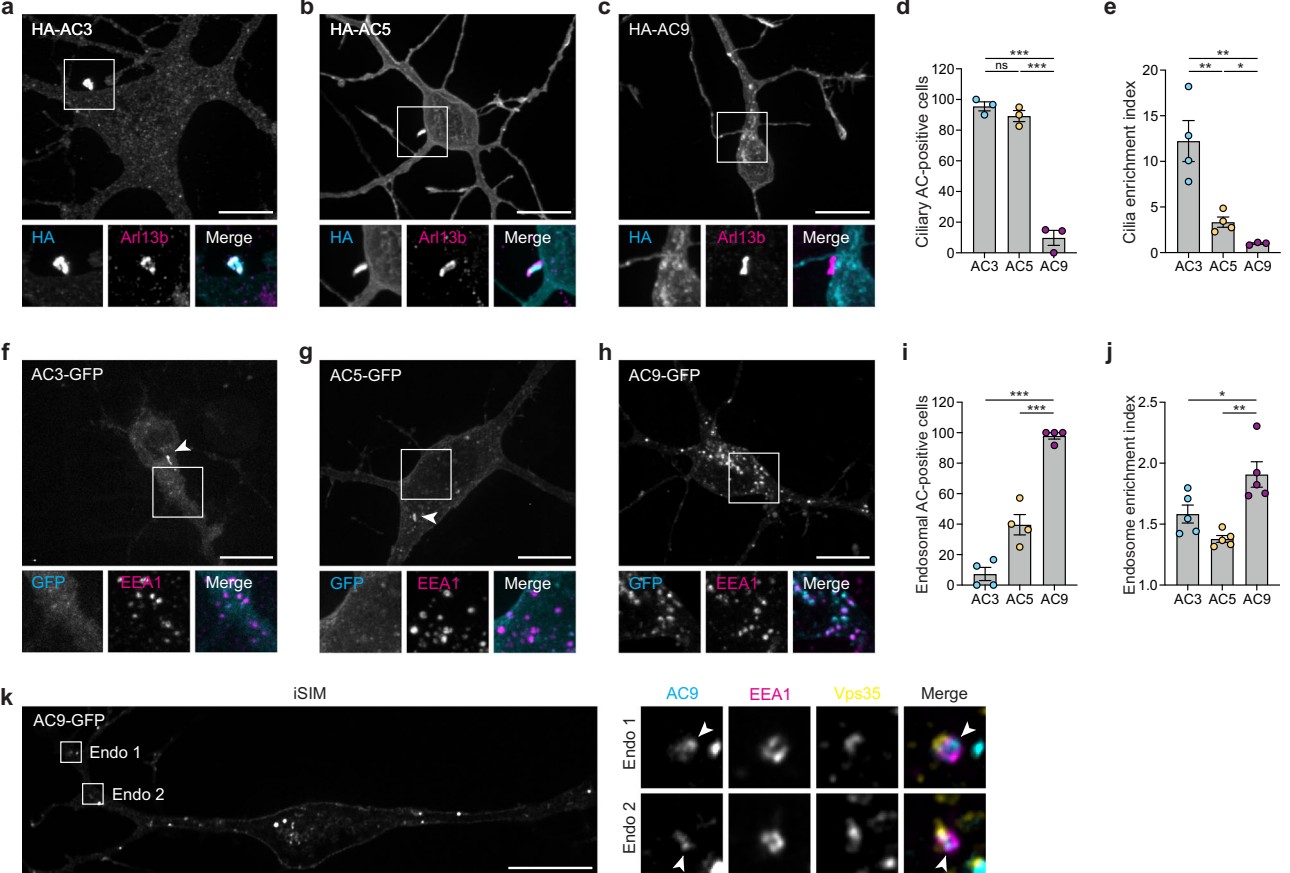

**Fig. 1 | Adenylyl cyclases isoforms localize to primary cilia and endosomes. a–c** Maximum intensity Z-projection of confocal microscopy images of MSNs expressing HA-AC3 (**a**), HA-AC5 (**b**), and HA-AC9 (**c**) and stained for cilia marker Arl13b. **d** Quantification of cilia localization by measuring the fraction of neurons with AC isoforms localized to cilia, using FLAG-D1R as cilia marker (not shown), and expressed as a percentage of total ciliated cells in the transfected cell population. Data are shown as mean ± s.e.m. from *n* = 3 independent experiments (total of 68–73 cells/condition). AC3 vs AC9, ***P* = 0.0001, AC5 vs. AC9, ***P* = 0.0002. **e** Cilia enrichment index calculated as a ratio of HA-AC fluorescence intensity in cilia (determined by Arl13b and FLAG-D1R) divided by total cell fluorescence. Data are shown as mean ± s.e.m. from *n* = 4 independent experiments (for AC3 and AC5) and *n* = 3 independent experiments (for AC9) (31–43 cells/condition). AC3 vs. AC9, ***P* = 0.0083, AC3 vs. AC5, ***P* = 0.0085, AC5 vs. AC9, **P* = 0.0175. **f–h** Representative confocal microscopy images of MSNs expressing AC3-GFP (**f**), AC5-GFP (**g**) and AC9-GFP (**h**) and stained for endosomal marker EEA1. Arrowheads indicate cilia. **i** Quantification of endosome localization by measuring the fraction of neurons with 10 or more AC-GFP internal puncta, expressed as a percentage of total transfected cells. Data are shown as mean ± s.e.m. from *n* = 4 independent experiments (34–47 cells total/condition). AC3 vs AC9, ***P* < 0.0001, AC5 vs. AC9, ***P* = 0.0002. **j** Endosome enrichment index calculated as a ratio of AC-GFP fluorescence intensity at EEA1-positive endosomes divided by total cell fluorescence. Data are shown as mean ± s.e.m. from *n* = 5 independent experiments; 38–58 cells total/condition. ***P* = 0.0012, **P* = 0.0351. **k** Representative iSIM images from *n* = 3 independent experiments of MSN expressing AC9-GFP and stained for EEA1 and Vps35. Arrowheads indicate AC9-positive subdomains at the endosomal membrane. Scale bars are 10 μm. n.s. not significant, *P* values are calculated using unpaired two-tailed Student's *t*-test. Source data are provided as a Source Data file.

also to intracellular puncta (Fig. 1c). Ciliary localization of HA-AC3 and HA-AC5 was observed in over 90% of neurons examined but was not detected with HA-AC9 (Fig. 1d). Fluorescence intensity analysis verified selective enrichment of both AC3 and AC5, but not AC9, in the ciliary microdomain, with AC3 concentrating more strongly than AC5 (Fig. 1e).

We confirmed these isoform-specific subcellular distribution patterns using AC constructs labeled in the C-terminus rather than N-terminus with a monomeric GFP variant[32] (muGFP) rather than an epitope tag (Fig. 1f–h, arrowheads in each image indicate the cilium). Interestingly, the vast majority of AC9-containing intracellular puncta overlapped with two established endosome markers, EEA1 (Fig. 1h, lower panels) and Vps35 (Supplementary Fig. 1a). Robust intracellular accumulation of AC9 was observed in almost all neurons examined (Fig. 1i) and we verified the selective endosomal enrichment of AC9 by fluorescence intensity measurement using EEA1 (Fig. 1j) or Vps35 (Supplementary Fig. 1b) to define an endosome mask. Using instant Structured Illumination Microscopy (iSIM) to obtain a higher spatial resolution, we found AC9-GFP to be concentrated on subdomains of the endosomal limiting membrane (Fig. 1k, arrowheads) and, in some endosomes, it appeared to be in the endosome lumen (Supplementary Fig. 1c). As only proteins present in the endosome limiting membrane would have the potential to generate cAMP in the cytoplasm, we specifically tested for the presence of AC9 in the limiting membrane by replacing GFP tag with supereclipic pHluorin (SEP), a GFP variant that is fluorescent when exposed to the cytoplasm but whose fluorescence is quenched in the acidic environment of the endosome lumen[33]. This strategy has been previously validated to selectively visualize proteins present in the endosomal limiting membrane[34]. AC9-SEP fluorescence

was indeed visible in EEA1-marked endosomes, and it resolved in a donut-like distribution characteristic of limiting membrane localization (Supplementary Fig. 1d), thereby confirming the presence of AC9 in the endosome limiting membrane.

Altogether, these data reveal a distinct subcellular distribution of AC isoforms in neurons: AC3 is highly concentrated in the ciliary microdomain. AC5 localizes on the soma both in cilia and on the extraciliary surface, extending into dendritic and axonal processes. AC9 localizes on the neuronal plasma membrane exclusively outside of cilia and is selectively concentrated in endosomes, where it appears to be organized in membrane microdomains.

## D1Rs colocalize with ACs at each isoform-selective membrane location

GPCR-mediated stimulation of AC activity is a membrane-delimited process thought to require the presence of activated receptors and ACs in the same bilayer[35]. MSNs endogenously express the dopamine 1 receptor (D1R), a $G\alpha_{s/olf}$-coupled GPCR whose activation can also occur at endosomes, Golgi, and primary cilia, resulting in distinct downstream outcomes[36–38]. Accordingly, we next investigated the subcellular localization of each isoform relative to D1R. Confirming previous results[39,40], FLAG-tagged D1R localized at the plasma membrane but also in the ciliary microdomain together with HA-AC3 (Fig. 2a) and HA-AC5 (Fig. 2b).

In contrast to D1R targeting to the primary cilium, which was evident in the absence of receptor activation, D1R localization in endosomes is stimulated by agonist and requires receptor delivery to endosomes by clathrin-mediated endocytosis[41,42]. We used live cell imaging to ask if D1Rs accumulate after activation in AC9-positive

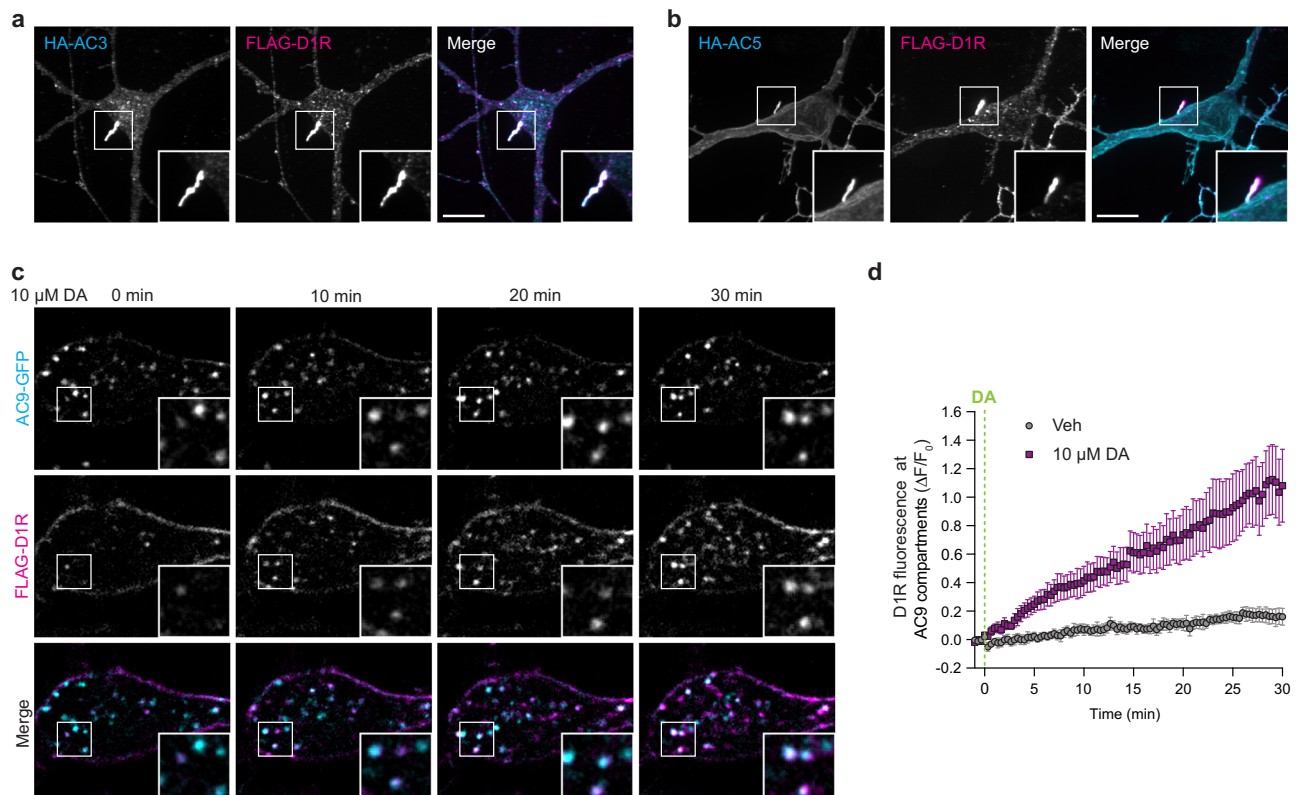

**Fig. 2 | Dopamine 1 receptor localizes to adenylyl cyclase isoforms positive compartments. a, b** Maximum intensity Z-projection of confocal microscopy images of MSNs expressing FLAG-D1R and HA-AC3 (**a**) or HA-AC5 (**b**) from $n = 4$ independent experiments. Scale bar = 10 μm. **c** Representative live cell spinning disk confocal images of MSNs transfected with AC9-GFP and FLAG-D1R and treated with 10 μM dopamine (DA) at 0 min. Surface FLAG-D1R was labeled with Alexa Fluor 555-coupled anti-FLAG antibody for 15 min before imaging. Scale bar = 5 μm. **d** Quantification of surface labeled FLAG-D1R accumulation at segmented AC9-GFP-positive endosomes after vehicle (Veh) or 10 μM DA addition. Data are shown as mean ± s.e.m. from $n = 5$ independent experiments; (33–35 cells total/condition). Source data are provided as a Source Data file.

endosomes. Surface-labeled FLAG-D1Rs redistributed from the plasma membrane to internal membrane puncta within minutes after addition of 10 μM dopamine to the culture medium. Further, the internalized D1Rs were found to accumulate in the same puncta containing AC9 (Fig. 2c) that were defined as endosomes by colocalization with EEA1 (Supplementary Fig. 2a, b). Fluorescence intensity analysis revealed that D1Rs begin to accumulate in AC9-containing endosomes within ~1 min after dopamine application and then progressively accumulate over a prolonged time course (Fig. 2d). The localization of AC9 to endosomes, in contrast, remained stable (Supplementary Fig. 2c, d). The β2-adrenergic receptor (β2AR), another GPCR capable of signaling from endosomes[43], also accumulated in endosomes stably containing AC9 following activation by the β-adrenergic receptor agonist iso-proterenol (Supplementary Fig. 2e, f).

Altogether, these results indicate that D1Rs can populate each of the subcellular membrane domains in which striatal AC isoforms distribute: D1R localizes in the plasma membrane, both to the ciliary microdomain containing AC3 and AC5 and to the extraciliary domain containing AC5 and AC9. D1R can also robustly accumulate in endosomes containing AC9, but this process is ligand-dependent.

## The N-terminal cytoplasmic domain harbors isoform-selective targeting determinants

Having established distinct subcellular localization patterns among the AC isoforms examined, we next investigated the structural basis for the isoform-selective targeting. Transmembrane ACs have a shared membrane topology and extensive sequence homology, but the N-terminal cytoplasmic domain of ACs is highly divergent[10]. Accordingly, we focused on this domain as a potential determinant of isoform-selective AC targeting.

To examine AC targeting to the primary cilium, we assessed the effect of replacing the N-terminus of AC5, which selectively concentrates in the primary cilium but is also detectable in the plasma membrane outside of the cilium, with the corresponding sequence derived from the non-ciliary isoform AC9 (Fig. 3a). We found that this chimeric mutant protein (AC5-AC9-Nter), while still observed in the plasma membrane, failed to localize on the primary cilium, labeled with FLAG-D1R (Fig. 3a, b). Deleting the N-terminus from AC5 (AC5-ΔNter) produced a similar phenotype (Fig. 3c, d). Together, these results indicate that the N-terminal cytoplasmic domain of AC5 is necessary for targeting this isoform to the cilium. Conversely, replacing the N-terminus of AC9 with the corresponding sequence derived from AC5 produced a chimeric mutant construct (AC9-AC5-Nter) that accumulated in the primary cilium (Fig. 3e, f). This indicates that the N-terminal cytoplasmic domain of AC5 is also sufficient to drive AC targeting to the cilium.

We noticed above that the chimeric mutant AC5-AC9-Nter (Fig. 4a) localized to intracellular puncta (Fig. 3a). To further investigate the structural determinants responsible for endosome targeting, we verified AC5-AC9-Nter accumulation in endosomes by colocalization with EEA1 (Fig. 4b). Using fluorescence intensity analysis, we determined that fusing the AC9 N-terminus to AC5 promotes endosomal concentration of the chimeric mutant to the same degree as observed for AC9 (Fig. 4c). These results indicate that the N-terminal cytoplasmic domain of AC9 is sufficient to drive isoform-selective AC targeting to endosomes.

## A dileucine motif in the AC9 N-terminus drives robust concentration in endosomes

We sought to map the endosomal targeting determinant in the AC9 N-terminal domain in more detail by dividing this sequence into three portions and examining the ability of each to drive chimeric AC5 targeting to endosomes (Fig. 4d). The middle portion of the AC9 N-terminal tail (amino acids 36–70) did not promote AC5 localization to endosomes, but both the proximal (amino acids 1–35) and distal

(amino acids 71–105) portions conferred detectable endosomal enrichment on AC5 chimeras (Fig. 4e). We noticed that the proximal portion contains a dileucine sequence (Fig. 4f, highlighted in red), which is known to have the potential to bind AP-2 and promote endocytosis via clathrin-coated pits[44]. Thus, we asked if the dileucine sequence present in the AC9 N-terminus acts as an endocytic determinant for this isoform. Mutating the dileucine residues to alanines within the AC5 chimera containing AC9 N-terminus proximal portion (AC5-AC9-1-35-LL > AA) led to a complete loss of endosomal localization (Fig. 4g, h). Furthermore, mutating this dileucine motif in full-length AC9 (AC9-LL > AA) significantly reduced AC9 enrichment in endosomes (Fig. 4i, j), causing a shift in the distribution of AC9 from endosomes to the plasma membrane. However, this mutation did not fully block visible AC9 targeting to endosomes, consistent with the distal portion of AC9 N-terminus also having endocytic activity.

The distal portion of the AC9 N-terminus did not contain any known endocytic motifs, so we divided this region into five overlapping sequences (Supplementary Fig. 3a) and assessed the ability of each to direct chimeric AC5 targeting to endosomes. While AC5-AC9-71-105 showed strong endosomal localization (Supplementary Fig. 3b), as observed above (Fig. 4e), all chimeric AC5 mutants showed decreased endosomal enrichment (Supplementary Fig. 3c–f, h), except for AC5-AC9-94-105 (Supplementary Fig. 3g, h) which showed no significant decrease. This suggests that the last 12 amino acids region (amino acids 94-105) of the AC9 N-terminus is important for endosomal localization. Further deletion of the first six amino acids of this region in the AC5-AC9-71-105 chimera (AC5-AC9-71-105ΔKFDSVN) or the last six amino acids (AC5-AC9-71-105ΔLEEACL) (Supplementary Fig. 3i–k) revealed that removing the LEEACL sequence resulted in a substantial loss of endosomal localization (Supplementary Fig. 3k). Deleting this sequence in full-length AC9 (Supplementary Fig. 3l, m) or mutating it into alanine (Supplementary Fig. 3n) led to decreased endosomal localization, confirming the requirement of this motif in promoting endosomal targeting. However, the targeting of these mutants to the plasma membrane was also affected, resulting in their accumulation in the endoplasmic reticulum (Supplementary Fig. 3l–n).

Altogether, these results indicate that the dileucine and the LEEACL sequences present in the proximal and distal segments of AC9 N-terminus, respectively, act as part of a bipartite endosomal targeting signal that drives the robust and specific endosomal concentration of AC9.

## Endosomal localization of AC9 impacts the timing of dopamine-induced PKA activity

We next investigated the significance of AC subcellular localization by assessing the impact of each isoform on the cAMP/PKA signaling cascade. AC3 and AC5 overexpression significantly elevated levels of both global cAMP concentration and PKA activity elicited by endogenous dopamine receptor activation (Fig. 5a, b). Surprisingly, AC9 overexpression did not detectably increase global cAMP, but it significantly augmented the elevation of PKA activity elicited by endogenous dopamine receptors (Fig. 5c).

To ask if the effect of AC9 on PKA activity but not global cAMP is related to its concentration in endosomes, we used the dileucine mutation (AC9-LL > AA) to selectively reduce AC9 endosomal localization (Fig. 4i, j). AC9 and AC9-LL > AA showed similar AC activity and were expressed at comparable levels (Supplementary Fig. 4a,b). Notably, the expression of AC9-LL > AA did not impact the overall endogenous dopamine-induced cAMP production when compared to AC9 expression. In both conditions, a rapid cAMP elevation occurred within 2 min and decayed over several minutes to a plateau above baseline in the continuous presence of agonist (Fig. 5d). Both the rapid initial elevation and the plateau phase remained unaltered by AC9-LL > AA expression (Fig. 5e, f). Subsequently, we evaluated the impact of AC9 mis-localization on endogenous dopamine-elicited elevation of global

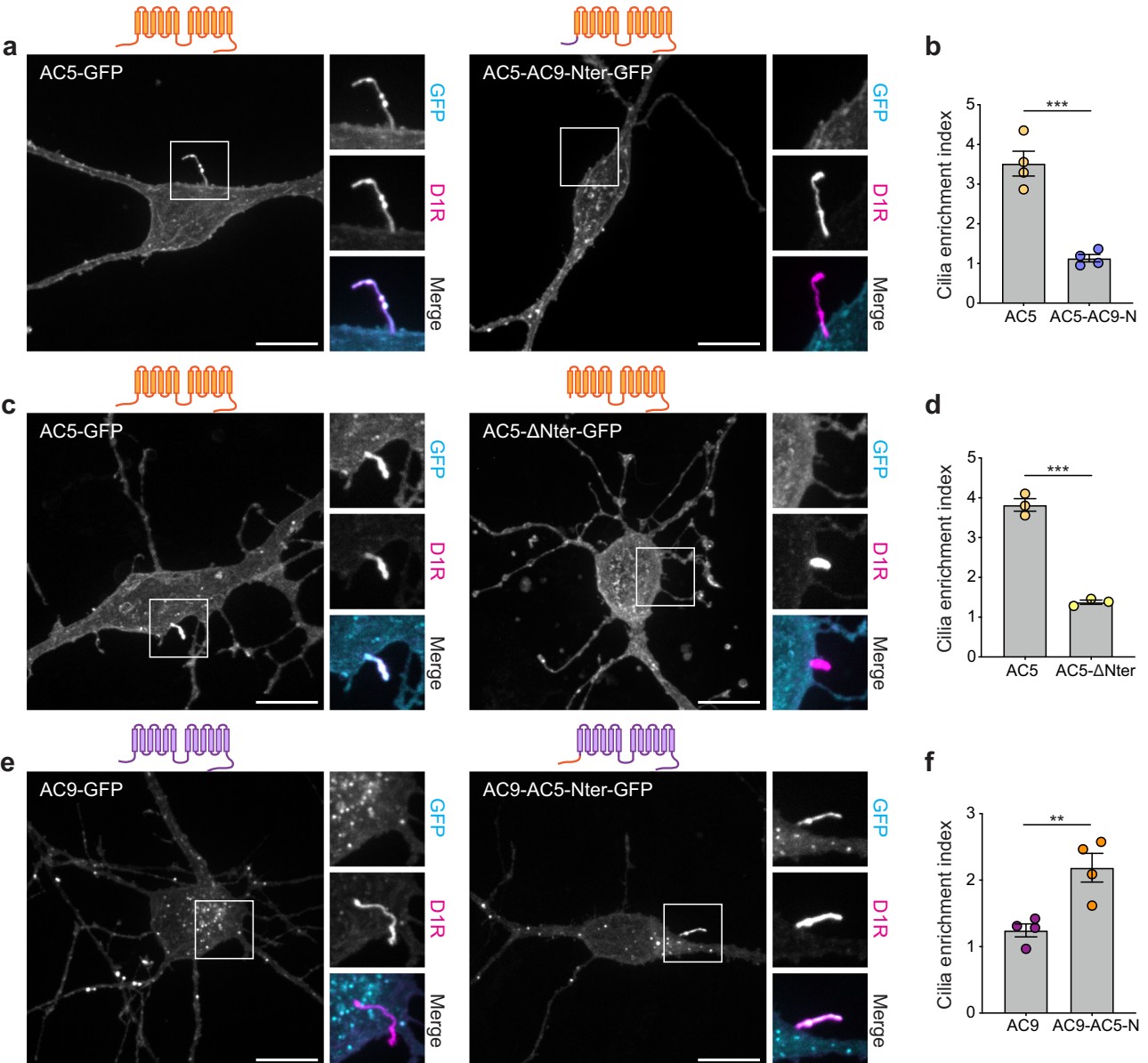

**Fig. 3 | The N-terminus of AC5 is necessary and sufficient for cilia targeting.**
**a, c, e** Maximum intensity Z-projection of confocal images microscopy images of MSNs surface-labeled for FLAG-D1R and expressing AC5-GFP or AC5-AC9-Nter-GFP (**a**), AC5-GFP or AC5-ΔNter-GFP (**c**), AC9-GFP or AC9-AC5-Nter-GFP (**e**). Show above each panel is the schematic representation of each construct showing corresponding mutations and topology. **b, d, f** Cilia enrichment index of cells coexpressing FLAG-D1R and AC5-GFP or AC5-AC9-Nter-GFP (**b**) (*n* = 4 independent experiments; 35-38 cells total/condition; ***P = 0.0003 by unpaired two-tailed Student's *t*-test), AC5-GFP or AC5-ΔNter-GFP (**d**) (*n* = 3 independent experiments; 39 cells total/condition; ***P = 0.0001 by unpaired two-tailed Student's *t*-test), AC9-GFP or AC9-AC5-Nter-GFP (**f**) (*n* = 4 independent experiments; 36 cells total/condition; **P = 0.0074 by unpaired two-tailed Student's *t*-test). Data are shown as mean ± s.e.m. Scale bars are 10 μm. Source data are provided as a Source Data file.

PKA activity using the ExRai-AKAR2 fluorescent biosensor[45]. Surprisingly, while the initial peak of PKA activity remained unchanged between the two conditions, AC9-LL > AA was significantly impaired relative to wild type AC9 in its ability to support the plateau phase of PKA activity elevation (Fig. 5g-i). Using an endosome targeted-ExRai-AKAR2 (ExRai-AKAR2-Endo) (Supplementary Fig. 5a), we similarly observed a decrease in the later phase of PKA activity at endosomes (Supplementary Fig. 5a–c), which is consistent with the AC9-LL > AA mutant showing less enrichment in endosomes compared to wild-type AC9 (Fig. 4i, j).

These observations suggest that the endosomal localization of AC9 impacts the timing of the PKA response and, specifically, the ability to maintain the later plateau phase of activity elevation. However, AC9 localization to endosomes did not detectably affect global cAMP levels, suggesting that the dopamine-stimulated cAMP

produced by AC9 at the endosomes represents only a small fraction of the total dopamine-stimulated cAMP production. The significant impact of these low levels of endosomal cAMP produced by AC9 on PKA activity led us to investigate in more detail the spatial relationship in neurons between AC9-containing endosomes and PKA.

## AC9-containing endosomes dynamically contact juxtanuclear neuronal PKA stores

PKA is a heterotetramer consisting of two catalytic subunits and two regulatory subunits, with PKA regulatory subunits IIβ (PKA RIIβ) being the main isoforms in the striatum[22,46–48]. Super-resolution iSIM microscopy analysis revealed close proximity and intertwining of AC9-positive endosomes with both PKA catalytic α (PKA cat) (Fig. 6a, b and Supplementary Movie 1) and PKA RIIβ (Fig. 6c, d and Supplementary

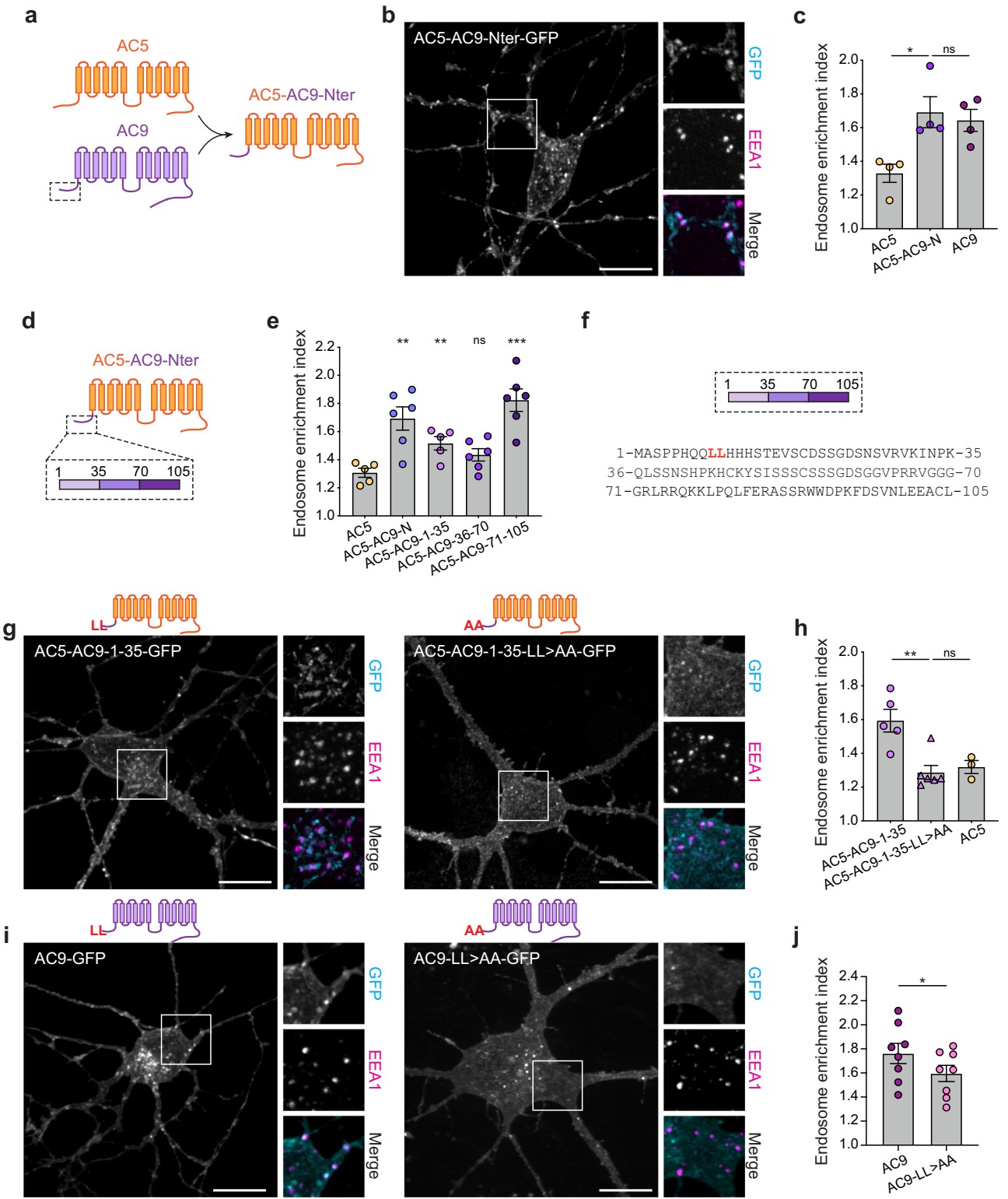

**f**

```
       1   35      70      105

1-MASPPHQQLLHHHSTEVSCDSSGDSNSVRVKINPK-35
36-QLSSNSHPKHCKYSISSSCSSSGDSGGVPRRVGGG-70
71-GRLRRQKKLPQLFERASSRWWDPKFDSVNLEEACL-105
```

Movie 2) compartments. These tubulovesicular PKA cat and RIIβ compartments were localized in the perinuclear region in a distribution overlapping that of Golgi membranes (Supplementary Fig. 6a, b), as previously observed[49,50]. Furthermore, we verified extensive colocalization between PKA cat and RIIβ in the Golgi region (Supplementary Fig. 6c), consistent with complex formation there. Further analyses confirmed significant contact between AC9-containing endosomes with PKA compartments, with ~30 AC9-positive puncta colocalizing with PKA RIIβ in the cell body (Supplementary Fig. 7a, b,

g). AC9-LL > AA showed fewer contacts (~20 puncta), although this difference was not statistically significant relative to AC9 WT (Supplementary Fig. 7c, d, g). This is consistent with the dileucine mutation reducing the degree of concentration in endosomes but not fully preventing endosome localization. In contrast, AC5 showed almost no colocalization (~3 puncta) (Supplementary Fig. 7e–g). Live cell microscopy of neurons revealed dynamic close interactions between AC9-positive endosomes and PKA cat puncta, following dopamine addition, with contact times ranging from several minutes up to ~30 min (Fig. 6e,

**Fig. 4 | A dileucine motif in the N-terminus of AC9 is required for endosome localization. a** Schematic representation of the chimeric mutant AC5-AC9-Nter. The AC5-derived sequence is depicted in orange and the AC9-derived sequence in purple. **b** Maximum intensity Z-projection of confocal images microscopy images of MSNs expressing AC5-AC9-Nter-GFP and stained for EEA1. **c** Endosome enrichment index of cells expressing AC5-GFP, AC5-AC9-Nter-GFP or AC9-GFP. Data are shown as mean ± s.e.m. from $n = 4$ independent experiments; 46-52 cells total/condition; *$P = 0.0144$ by unpaired two-tailed Student's $t$-test. **d** Schematic of chimeric mutants of AC5 (in orange) containing portions of AC9 N-terminus (in purple). **e** Endosome enrichment of cells expressing AC5 chimeric mutants. Data are shown as mean ± s.e.m. from $n = 5$ independent experiments for AC5 and AC5-AC9-1-35 and $n = 6$ independent experiments for AC5-AC9-N, AC5-AC9-36-70 and AC5-AC9-71-105 (≥26 cells total/condition). ***$P = 0.0004$, **$P = 0.003$ for AC5 vs. AC5-

AC9-N, **$P = 0.0067$ for AC5 vs. AC5-AC9-1-35 by unpaired two-tailed Student's $t$-test. **f** Schematic of the three portions of AC9 N-terminus and their respective sequence. **g** Representative confocal images of MSNs expressing AC5-AC9-1-35-GFP or AC5-AC9-1-35-LL > AA-GFP and stained for EEA1. **h** Endosome enrichment of cells in **g**. Data are shown as mean ± s.e.m. from $n = 3$ independent experiments for AC5, $n = 5$ for AC5-AC9-1-35 and $n = 6$ for AC5-AC9-1-35-LL > AA (≥30 cells total/condition). **$P = 0.003$ by unpaired two-tailed Student's $t$-test. **i** Representative confocal images of MSNs expressing AC5-AC9-1-35-GFP or AC5-AC9-1-35-LL > AA-GFP and stained for EEA1. **j** Endosome enrichment of cells in **i**. Data are shown as mean ± s.e.m from $n = 8$ independent experiments (69–72 cells total/condition). *$P = 0.0406$ by paired two-tailed Student's $t$-test. Scale bars are 10 μm. Source data are provided as a Source Data file.

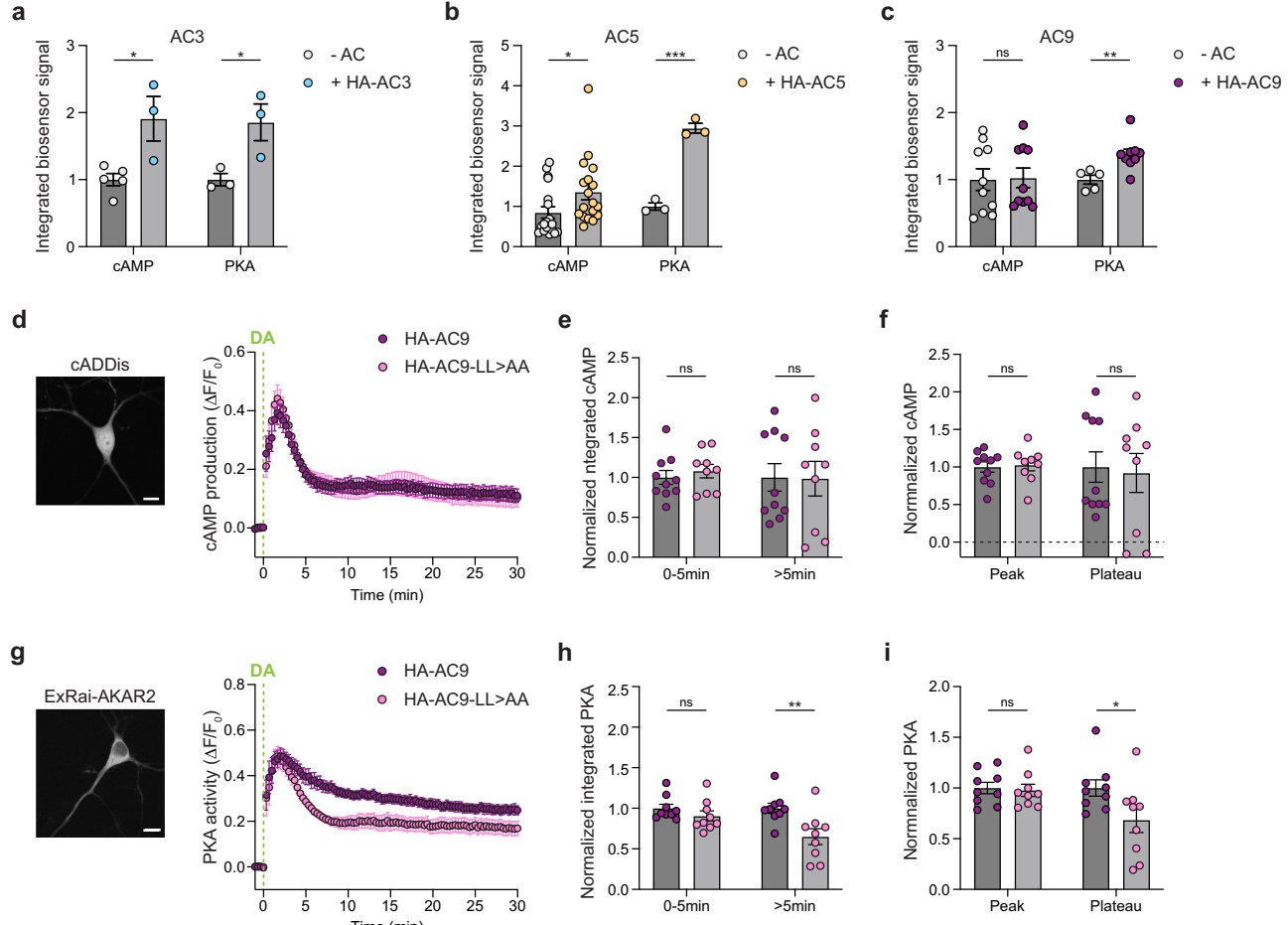

**Fig. 5 | Endosomal AC9 modulates PKA activity but not overall cAMP. a–c** cAMP and PKA response in cells expressing cADDis (cAMP) or ExRai-AKAR2 (PKA) sensors alone or with HA-AC3 (**a**), HA-AC5 (**b**), or HA-AC9 (**c**) and treated with 10 μM dopamine (DA). **a** $n = 5$ and 3 independent experiments (cAMP, -AC and HA-AC3, respectively; 60-64 cells total/condition), and $n = 3$ (PKA; 42-61 cells total/condition). **b** $n = 18$ (cAMP; 293-310 cells total/condition), $n = 3$ (PKA; 28-42 cells total/condition). **c** $n = 10$ (cAMP; 128-189 cells total/condition), $n = 5$ and 9 (PKA, -AC and HA-AC9 respectively; 78–126 cells total/condition). **a** *$P = 0.0154$ for cAMP and *$P = 0.0412$ for PKA. **b** ***$P = 0.0002$, *$P = 0.0432$. **c** **$P = 0.0072$. **d** Representative image of MSN expressing the cAMP biosensor cADDis and kinetics of cAMP production over time in MSNs coexpressing cADDis and HA-AC9 (purple, $n = 10$) or HA-AC9-LL > AA (pink, $n = 9$) (173–189 cells total/condition) and treated with 10 μM DA. **e** Integrated cAMP signals of the phases 0-5 min and >5 min (5–30 min) after DA

addition ($n = 9$ for HA-AC9, $n = 10$ for HA-AC9-LL > AA). **f** Peak and plateau values were calculated as the maximum $\Delta F/F_0$ (peak) and the average of 20-3$_0$ min values (plateau) ($n = 9$ for HA-AC9, $n = 10$ for HA-AC9-LL > AA). **g** Representative image of MSN expressing the PKA biosensor ExRai-AKAR2 and kinetics of PKA activity over time in MSNs coexpressing ExRai-AKAR2 and HA-AC9 (purple) or HA-AC9-LL > AA (pink) and treated with 10 μM DA, from $n = 9$ independent experiments (120–126 cells total/condition). **h** Integrated PKA signals of the phases 0-5 min and >5 min (5–30 min) after DA addition ($n = 9$ independent experiments; **$P = 0.0082$). **i** Peak and plateau values ($n = 9$ independent experiments; *$P = 0.0483$). For all panels, data represent biological replicates and are shown as mean ± s.e.m. Scale bars are 10 μm. $P$ values are calculated by unpaired two-tailed Student's $t$-test. Source data are provided as a Source Data file.

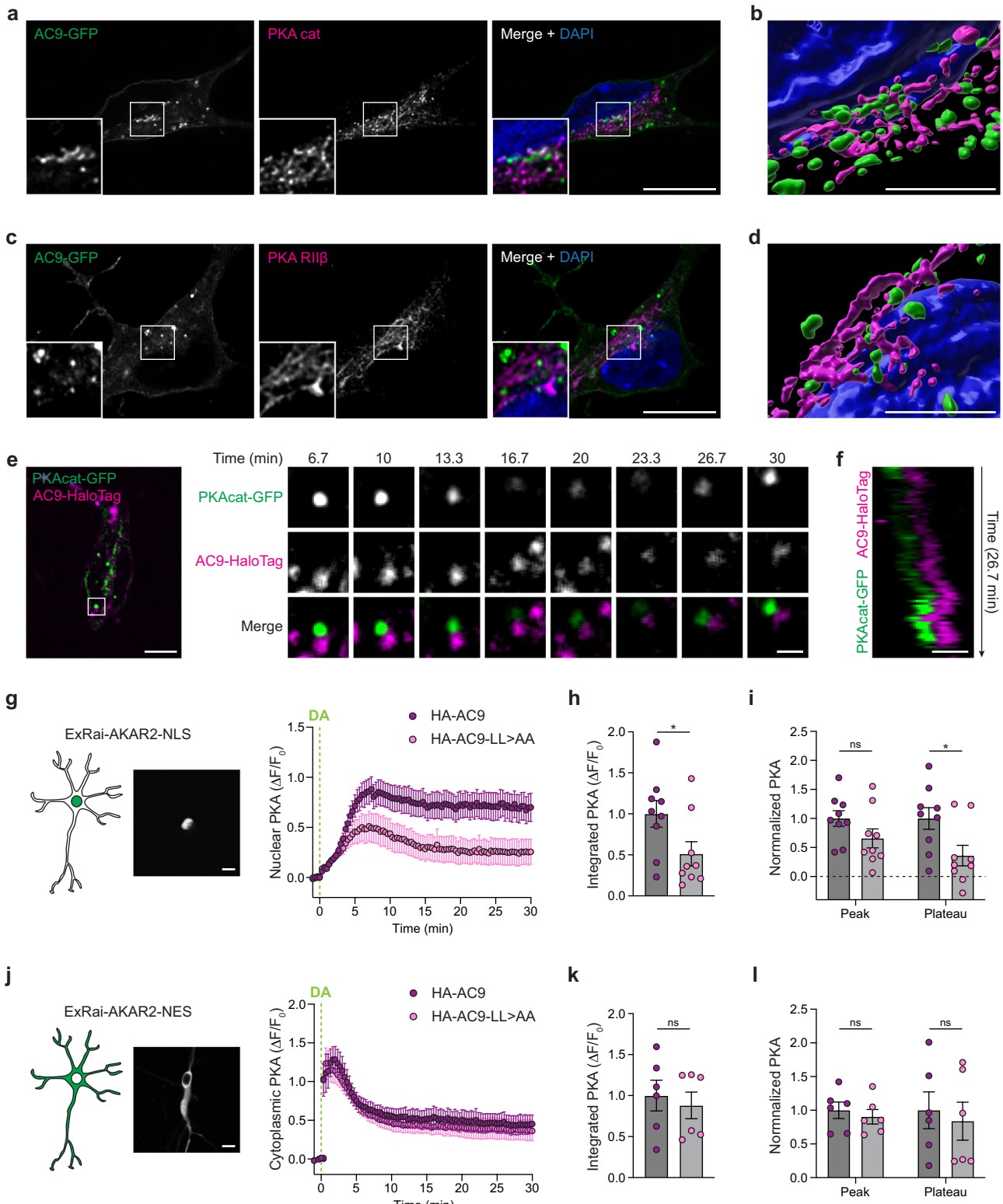

Supplementary Movie 3 and Supplementary Fig. 8). Close contact between AC9-positive endosomes and perinuclear PKA cat stores was suggested by coordinated movement of the two adjacent proteins in sequential image series (Fig. 6f).

## Concentration in endosomes enables AC9 to specifically control nuclear PKA activity

A prevailing current view of how cAMP elevates nuclear PKA activity is by inducing dissociation of catalytic subunits from regulatory subunits, disinhibiting the kinase activity of catalytic subunits and enabling a fraction of them to diffuse into the nucleus[51–53]. Consistent with this, the PKA biosensor ExRai-AKAR2 detected increased activity in the cytoplasm, followed by the nucleus (Supplementary Movie 4). Based on the close apposition between AC9 endosomes and PKA compartments in the perinuclear region, we wondered whether the ability of AC9 endosomal localization to increase PKA activity could reflect an effect of PKA activity elevation in the nucleus. To test this, we generated a nucleus localized-ExRai-AKAR2 (ExRai-AKAR2-NLS) and

**Fig. 6 | Nuclear PKA activity is dependent on AC9 endosomal concentration.**
**a, c** Representative iSIM images of MSN expressing AC9-GFP and stained for endogenous PKA cat (**a**, $n = 4$ independent experiments) or PKA RIIβ (**c**, $n = 3$). Nuclei were stained with DAPI. Scale bar = 10 μm. **b,d**, 3D rendering of cells in **a** (**b**) and **c** (**d**). Scale bar = 5 μm. **e** Spinning-disk confocal images from a time series of neurons coexpressing PKAcat-GFP and AC9-HaloTag and treated with 10 μM dopamine (DA) at $t = 0$ min, from $n = 3$ independent experiments. Scale bar = 5 μm. **f** Kymograph of the cell in (**e**). Scale bar = 1 μm. **g** On the left, schematic representation and spinning-disk confocal representative image showing ExRai-AKAR2-NLS localization to the nucleus (Scale bar = 10 μm). On the right, kinetics of nuclear PKA activity over time in MSNs coexpressing ExRai-AKAR2-NLS and HA-AC9 (purple) or HA-AC9-LL > AA (pink) and treated with 10 μM DA, from $n = 9$ independent experiments (61–69 cells total/condition). **h** Integrated nuclear PKA signal ($n = 9$

independent experiments; *$P = 0.0409$ by unpaired two-tailed Student's $t$-test).
**i** Peak and plateau values were calculated as the maximum $\Delta F/F0$ (peak) and the average of 20–30 min values (plateau) ($n = 9$ independent experiments; *$P = 0.0242$ by unpaired two-tailed Student's $t$-test). **j** On the left, schematic representation and spinning-disk confocal representative image showing ExRai-AKAR2-NES localization to the cytoplasm and excluded from the nucleus (Scale bar = 10 μm). On the right, kinetics of cytoplasmic PKA activity over time in MSNs coexpressing ExRai-AKAR2-NES and HA-AC9 (purple) or HA-AC9-LL > AA (pink) and treated with 10 μM DA, from $n = 6$ independent experiments (61–103 cells total/condition).
**k** Integrated cytoplasmic PKA signal ($n = 6$ independent experiments). **l** Peak and plateau values ($n = 6$ independent experiments). For all panels, data represent biological replicates and are shown as mean ± s.e.m. Source data are provided as a Source Data file.

co-expressed it with AC9 or AC9-LL > AA in neurons (Fig. 6g). Dopamine addition elicited a gradual increase in nuclear PKA activity that peaked between 5 and 10 min, followed by a plateau phase (Fig. 6g). Both the total nuclear PKA activity (Fig. 6h) and the plateau phase were strongly reduced upon AC9-LL > AA expression (Fig. 6i). As an alternative approach, we measured nuclear PKA activity using the non-targeted ExRai-AKAR2 (Fig. 5g) by drawing a ROI within the nucleus and measuring fluorescence over time (Supplementary Fig. 9a). AC9-LL > AA expression similarly resulted in a significant reduction in nuclear PKA activity (Supplementary Fig. 9b,c), affecting both the peak and plateau phases (Supplementary Fig. 9d). These results suggest a critical role for endosomal concentration of AC9 in promoting the ability of endogenous dopamine receptor activation to elevate PKA activity in the nucleus.

We next carried out the converse analysis, constructing an ExRai-AKAR2 variant that is specifically localized in the cytoplasm and excluded from the nucleus (ExRai-AKAR2-NES, Fig. 6j). Remarkably, and in contrast to the clear difference observed for PKA activity elevation in the nucleus, reducing AC9 concentration in endosomes (AC9-LL > AA) produced no discernible effect on the dopamine-elicited elevation of PKA activity in the cytoplasm (Fig. 6k, l). Altogether, our data collectively reveal that AC9 endosomal localization specifically and uniquely modulates PKA activity by promoting PKA activity elevation in the nucleus, without detectably affecting cytoplasmic activity.

## Discussion

Local cAMP signaling from any subcellular location requires cAMP to be locally produced, but a key knowledge gap in our present understanding is how ACs are localized. We addressed this by focusing on striatal MSNs, a native cell type in which the cAMP/PKA cascade mediates most of the physiological actions of dopamine[54]. We describe a precise subcellular organization of striatum-expressed AC isoforms in these neurons, delineate a discrete cellular mechanism for targeting AC9 to endosomes, and reveal an elaborate membrane network in the somatic cytoplasm that enforces proximity between AC9 and PKA in the juxtanuclear cytoplasm. We also show that the efficient sorting of AC9 into this network provides a selective way for AC9 to regulate PKA activity in the nucleus.

AC3 has been previously documented to localize to the primary cilium, a restricted microdomain of the somatic plasma membrane[28,31]. Our results verify this and reveal additional diversity among isoforms in surface distribution. We show that AC5 also concentrates in the ciliary microdomain, but unlike AC3, it is enriched to a reduced degree and is also present on the extraciliary surface of the soma and dendrites. We then identify the divergent AC N-terminal cytoplasmic domain as a key structural determinant of isoform-selective differences in AC targeting. AC9 differs from both AC3 and AC5 in being excluded from the ciliary microdomain and specifically concentrated in endosomes. We previously showed AC9 localization to endosomes in HEK293 cells, but its trafficking to endosomes relies on ligand-

induced activation of the β2-adrenergic receptor and is sensitive to additional environmental cues that remain poorly defined[55]. Here we demonstrate a robust and consistent endosomal localization in relevant cell types that is not dependent on receptor activation. Our data indicate that this difference in AC9 trafficking is cell type-specific rather than GPCR-specific, as AC9 was observed in endosomes constitutively in MSNs, and this endosomal localization was not detectably affected by activation of either the D1R or β2-AR. We do not know the mechanistic basis for this cell type-specific difference in AC9 trafficking and suggest this as a future direction of study. We define a specific dileucine sequence within the proximal region of the AC9 N-terminus that is necessary for robust endosomal concentration of AC9 but show that the AC9 N-terminus contains an additional determinant contributing to endosomal enrichment that remains to be more fully defined. We note that each of the differentially localized striatal AC isoforms represents a distinct AC subclass, as defined by differential sensitivity to or regulation by $G_{i/o}$ and calcium: AC3 is weakly activated by $Ca^{2+}$/calmodulin[56], AC5 is inhibited by $G\alpha_i$ and $Ca^{2+}$[57] and AC9 is insensitive to both[58]. Accordingly, our results delineate a precise spatial landscape of the AC system that places AC isoforms differing in biochemical regulation at distinct subcellular membrane locations (Fig. 7a).

Here, we establish a functional significance of AC sorting in the spatial landscape by focusing on AC9's unique and specific concentration in endosomes. Previous studies have shown that local cAMP production from internal membranes can confer advantages over the plasma membrane in facilitating signaling in the nucleus[59–64]. Our results are fully consistent with this concept and advance it in several ways. Our study identifies AC9 as a pivotal factor responsible for local cAMP signaling from endosomes, while also uncovering the mechanisms governing AC9 targeting to endosomes. We show that AC9-containing endosomes are closely apposed to juxtanuclear PKA stores by being intimately intercalated with Golgi membranes on which type II PKA is bound. We have focused on PKA cat α and PKA RIIβ isoforms but note that additional isoforms have been shown to be expressed in the brain and are differently localized[47,65]. We further show that concentration in this intercalated membrane network enables AC9 to selectively elevate PKA activity in the nucleus relative to the cytoplasm. The present results also provide evidence supporting the ability of AC9 to locally activate PKA in proximity to AC9-containing endosomes. We suggest that future studies measuring cAMP gradients directly at a nanometer scale[2] will further extend our understanding of local cAMP signaling from endosomes.

The present results emphasize the elaborately intertwined nature of AC9-containing endosomes and Golgi-associated PKA compartments, characterized by extensive regions of close apposition. Local signaling by a diffusible mediator ultimately depends on the principle of enforced proximity[66]. Previous studies have established this principle in cAMP signaling, based on the lateral partitioning of AC and its molecular scaffolding with PKA[13]. This represents an *in cis* form of enforced proximity, based on proximity occurring on the

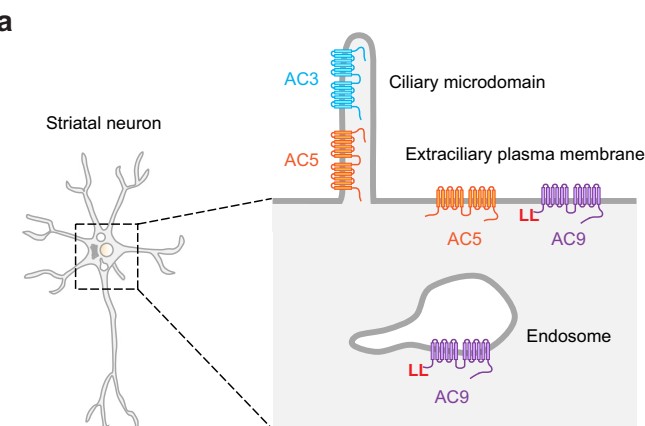
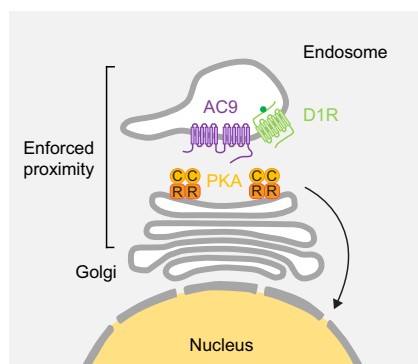

**Fig. 7 | Spatial organization of adenylyl cyclase isoforms impacts nuclear PKA activity in neurons. a** Diagram of proposed model illustrating the distinct sub-cellular distribution of AC isoforms in striatal neurons. AC3 is restricted to the primary cilium; AC5 localizes at the plasma membrane, both in cilia and extraciliary surfaces; AC9 localizes at the plasma membrane outside of cilia, and in endosomes.

Endosomal localization requires a dileucine motif in the AC9 N-terminus. **b** Endosomes containing AC9 and D1R are in close proximity with Golgi-associated PKA compartments in the perinuclear region. This enforced proximity 'in trans' selectively drives PKA activity in the nucleus (yellow).

same membrane surface. We propose that the present results reveal a discrete cellular mechanism for achieving enforced proximity in trans, based on proximity across adjacent membrane surfaces (i.e., endosome and Golgi membranes). The utility of membrane-membrane apposition or contact in achieving enforced proximity in cellular signal transduction is increasingly recognized, such as in signaling by calcium, lipids and reactive oxygen species[67]. The present results extend this general principle to cellular signaling by the cAMP cascade (Fig. 7b). Further studies will be needed to determine the biochemical basis for such close apposition between endosomal AC9 and Golgi-associated PKA.

Finally, our study reveals the unique ability of AC9 to selectively promote PKA elevation in the nucleus relative to the cytoplasm. Such specificity carries interesting physiological implications, as PKA activity within the nucleus has been linked to long-term cellular changes and learning processes[52]. Therefore, the ability of AC9 to selectively modulate nuclear PKA activity has the potential to significantly impact neuronal function and behavior. It is worth noting that AC9 is widely expressed in the brain and is particularly abundant in the hippocampus, a region associated with learning and memory[68]. Investigating whether AC9 endosomal localization extends to other neuronal cell types and whether it contributes to long-term plasticity associated with learning and memory would be of considerable interest. While the present study focused specifically on functional consequences of AC9 localization to endosomes, we anticipate that a similar experimental approach could be used to investigate the significance of subcellular AC localization more broadly, such as the effect(s) of AC3 and AC5 targeting to the primary cilium in specifying neuronal dopamine signaling.

## Methods
### Cell culture
Medium spiny neurons were prepared from embryonic day 18 Sprague-Dawley rats. After euthanasia of the pregnant Sprague-Dawley rat ($CO_2$ and bilateral thoracotomy), the brains of embryonic day 18 rats of both sexes were extracted from the skull. The striatum, including the caudate-putamen and nucleus accumbens, was dissected in ice-cold HBSS calcium/magnesium/phenol red-free (Thermo Fisher). Structures were dissociated in 0.05% trypsin/EDTA (UCSF Media

Production) for 15 min at 37 °C and washed in DMEM (Thermo Fisher) supplemented with 10% fetal bovine serum (UCSF Media Production) and 30 mM HEPES. Cells were then mechanically separated with a flame-polished Pasteur pipette and were plated onto poly-D-lysine coated 35 mm glass bottom dishes (Cellvis) in DMEM supplemented with 10% fetal bovine serum. Medium was exchanged on DIV 4-5 for phenol-free Neurobasal medium (Thermo Fisher) supplemented with GlutaMAX 1x (Thermo Fisher) and Gibco B-27 1× (Thermo Fisher) supplements. Half of the culture medium was exchanged every week with fresh, equilibrated medium. Cytosine arabinosine 2 µM (Millipore Sigma) was added at DIV 8. Transfection using Lipofectamine 2000 (Thermo Fisher) was performed at DIV 8 using 2 µl of Lipofectamine and 1–2 µg DNA in 1 ml of media per 35 mm imaging dish, and media was exchanged 4–6 h later. Cells were maintained in a humidified incubator with 5% $CO_2$ at 37 °C and imaged at DIV 11–15. All procedures were performed according to the National Institutes of Health Guide for Care and Use of Laboratory Animals and approved by the UCSF Institutional Animal Care and Use Committee (protocol number AN185688). HEK293 (human, CRL-1573) cells were maintained in DMEM containing 10% fetal bovine serum with 5% $CO_2$.

### Reagents and antibodies
Information on all chemicals and antibodies used in this study can be found in Supplementary Tables 1 and 2.

### DNA constructs
Information on all plasmids used in this study can be found in Supplementary Table 3.

N-terminally HA-tagged ACs were generated by PCR amplifying Homo sapiens AC3 cDNA (purchased from DNASU, HsCD00403688), Homo sapiens AC5-EGFP (previously generated in the von Zastrow lab by G. Peng from AC5 cDNA purchased from DNASU, HsCD00732284) and Homo sapiens AC9-EGFP (previously described in ref. [62]) with N-terminal HA tag added into primers followed by insertion into NotI/EcoRI-linearized pGCGFP-G418 (a gift from Andrew Pierce, Addgene plasmid #31264) using In-Fusion HD (Takara Bio). ACs C-terminally tagged with muGFP[32] were generated by removing EGFP from pEGFP-N1 vector by digestion with AgeI and NotI followed by PCR amplification of muGFP (a gift from Mullins lab) and insertion using In-Fusion

HD. Next, the generated pmuGFP-N1 vector was linearized with SacI and ApaI and PCR amplified AC3, AC5 and AC9 were inserted using In-Fusion HD. AC5-AC9-Nter-GFP was generated by PCR amplification of AC9 (residues M1-L105) and AC5 (D186-S1261) and insertion into pmuGFP-N1 using In-Fusion HD. AC5-ΔNter-GFP was generated by PCR amplification of AC5 (D192-S1261) and insertion into pmuGFP-N1 using In-Fusion HD. AC9-AC5-Nter-GFP was generated by PCR amplification of AC5 (M1-G185) and AC9 (residues E106-V1353) followed by insertion into pmuGFP-N1 using In-Fusion HD. AC5-AC9-1-35-GFP, AC5-AC9-36-70-GFP and AC5-AC9-71-105-GFP were generated by PCR amplification of AC9 (residues M1-K35, Q36-G70 and G71-L105, respectively) and AC5 (D186-S1261) and insertion into pmuGFP-N1 using In-Fusion HD. To generate AC5-AC9-1-35-LL > AA-GFP, a fragment of AC9 (residues M1-K35) containing the point mutations L9A and L10A (LL > AA) was generated by synthesis (Integrated DNA Technologies, IDT), followed by PCR amplification of AC5 (D192-S1261) and insertion into pmuGFP-N1. AC9-LL > AA-GFP was generated by PCR amplification of AC5-AC9-1-35-LL > AA-GFP (M1-K35) and AC9 (Q36-V1353) and insertion into pmuGFP-N1. For AC5-AC9-71-81-GFP, AC5-AC9-76-87-GFP, AC5-AC9-82-93-GFP, AC5-AC9-88-99-GFP, AC5-AC9-94-105-GFP, AC5-AC9-71-105-ΔKFDSVN-GFP and AC5-AC9-71-105-ΔLEEACL-GFP, corresponding fragments of AC9-Nterminus were generated by synthesis (Integrated DNA Technologies, IDT or Twist Bioscience), followed by PCR amplification of AC5 (D192-S1261) and insertion into pmuGFP-N1. For AC9-ΔLEEACL-GFP and AC9-LEEACL > AAAAAA-GFP, corresponding fragments of AC9-Nterminus were generated by synthesis (Twist Bioscience), followed by PCR amplification of AC9 (P148-V1353) and insertion into pmuGFP-N1. AC9-SEP was generated by PCR amplification of AC9 and SEP, followed by insertion into pGCGFP-G418. pcDNA3.1(+)-ExRai-AKAR2 (a gift from Jin Zhang, Addgene plasmid #161753) was subcloned into pGCGFP-G418. ExRai-AKAR2-Endo was generated by adding the FYVE domain of endofin at the C-terminus of ExRai-AKAR2. ExRai-AKAR2-NLS and -NES were previously generated in the von Zastrow lab by G. Peng by adding at the C-terminus 2x NLS motifs (SV40, 2x PKKKRKV) or a NES motif (LPPLERLTL), respectively, and subcloned into pGCGFP-G418. AC9-HaloTag was generated by PCR amplification of HaloTag and AC9, followed by insertion into pGCGFP-G418. PKAcat-GFP was previously generated in the von Zastrow lab by A. Marley by the addition of EGFP at the C-terminus and cloning into pCAGGS. FLAG-D1R and PKA RIIβ-mCh were previously generated in the von Zastrow lab by A. Ehrlich and G. Peng, respectively. FLAG-β2AR[62] was PCR amplified and inserted into pGCGFP-G418. All constructs were validated by DNA sequencing.

## Immunostaining, confocal and iSIM imaging

Neurons were fixed on DIV 11–12 with 4% paraformaldehyde and 4% sucrose for 10 min at room temperature and then washed 3 times with PBS. Cells were permeabilized with 0.1% Triton X-100 diluted in PBS for 5 min and washed 3 times with PBS. Cells were blocked in 1% BSA diluted in PBS for 30 min. Primary antibodies were diluted into a blocking solution and incubated for 30 min at room temperature. Neurons were washed 3 times with the blocking solution and incubated with secondary antibodies diluted in the blocking solution. Cells were washed 3 times in blocking solution, labeled with DAPI (Sigma-Aldrich) in PBS for 5 min when indicated, and washed 3 times with PBS. Neurons were imaged by confocal microscopy using a Nikon Ti inverted microscope equipped with a Yokogawa CSU-22 spinning disk unit, a Photometrics Evolve Delta EMCCD camera controlled by NIS-Elements 5.21.03 software and 488, 561, and 640 nm Coherent OBIS lasers. Samples were imaged using an Apo TIRF 100x/1.49 NA oil objective (Nikon), and 0.3 μm z-stacks were acquired. The iSIM was performed with a VT-iSIM super-resolution module on a Nikon Ti-E inverted microscope equipped with a 100×1.45NA Plan Apo objective and 405, 488, 561, and 647 nm lasers. Images were captured using a Hamamatsu Quest camera controlled by μManager 2.1 software

(https://www.micro-manager.org), with 0.25 μm z-stacks. Images were deconvoluted using Microvolution plugin on ImageJ Fiji[69]. Imaris software 10.1 was used to create 3D renderings.

## Cilia localization quantification

To measure the percentage of neurons with ACs in cilia, we imaged ciliated FLAG-D1R-expressing cells and counted the number of cells with visible HA-AC immunoreactivity on the cilium. The cilia enrichment index was measured on unprocessed maximum intensity Z-projection images using ImageJ Fiji[69]. Briefly, regions of interest (ROIs) were manually drawn around the cilium and the transfected cell on the FLAG-D1R or Arl13b channel. Fluorescence intensity of the HA-AC channel was measured inside both ROIs and background subtracted. The Cilia enrichment index was determined by dividing the cilia fluorescence by total cell fluorescence.

## Endosome localization quantification

To measure the percentage of neurons with AC-GFP in intracellular puncta, image analysis was performed on unprocessed images on ImageJ Fiji[69]. Maximum-intensity Z-projection images of cells transfected with AC-GFP were created. Then, internal puncta were identified by fluorescence values above a threshold set manually and the number of internal particles was quantified using the Analyze Particles command. Cells with 10 or more particles were considered positive, and the number of positive neurons was then divided by the total number of cells. To measure endosomal enrichment, image analysis was performed on unprocessed images using custom written code 'Endosome Enrichment' in MATLAB (Mathworks, R2020a). Maximum intensity Z-projection images of the AC-GFP channel and the endosomal marker (EEA1/Vps35) channel were created, and an ROI was drawn around the transfected cell on the AC channel. Endosomes within this region were defined by EEA1 or Vps35 fluorescence values above a threshold set manually. AC fluorescence intensity within the created endosomal mask and in the ROI were then measured and background subtracted. Endosomal enrichment index was determined by dividing the endosomal fluorescence by total cell fluorescence.

## Live cell imaging

Neurons were transfected on DIV 8 with Lipofectamine 2000. On the day of imaging, neurons expressing AC9-HaloTag were labeled with 200 nM JF$_{549}$ HaloTag ligand[70] for 15 min, then washed three times with pre-equilibrated HEPES buffered saline (HBS) solution (120 mM NaCl, 2 mM KCl, 2 mM MgCl$_2$, 2 mM CaCl$_2$, 5 mM glucose, 10 mM HEPES adjusted to pH 7.4). Neurons were incubated with HBS for 30 min and washed three times. Live cell maging was performed on a Nikon Ti inverted microscope equipped with an Andor Borealis CSU-W1 spinning disk confocal with solid-state 488, 561 and 640 nm lasers (Andor), Plan Fluor VC ×40 1.3 NA and Plan Apo VC ×100 1.4 NA objectives (Nikon) and an Andor Zyla 4.2 sCMOS camera controlled by μManager 2.0 software (https://www.micro-manager.org). Cells were kept at 37 °C in a temperature- and humidity-controlled chamber (Okolab). 10 μM dopamine was added after a 1 min baseline, and images were taken every 20 s for 30 min. Images displayed in Fig. 2c, Supplementary Fig. 2a, c, Fig. 6c, d and Supplementary Fig. 4 were processed on ImageJ Fiji[69] using the Subtract Background and Smooth commands.

## Receptor accumulation into endosomes

Image analysis was performed on unprocessed images using custom-written code 'Puncti' in MATLAB (Mathworks, R2020a). For quantifying receptor accumulation at endosomes, an ROI was drawn around the cell on the receptor channel, refined by thresholding a maximal temporal projection and smoothened. Endosomes within this region were defined by AC9-GFP (Fig. 2d) or DsRed-EEA1 (Supplementary Fig. 2b, d) fluorescence values above a threshold set manually. Filtering to exclude regions smaller than 3 pixels was applied, and the remaining

objects were smoothened with a dilatation of 1 pixel before an erosion of 1 pixel. The generated mask was applied to the D1R or AC9-GFP channel to measure the fluorescence at the location of marker objects. The average fluorescence in the mask was background subtracted and normalized to baseline (before agonist addition).

## Number of puncta colocalizing with PKA

3D renderings of AC puncta and PKA RIIβ were blindly created using the "Spot" and "Surface" features in Imaris software 10.1, respectively. The distance between each spot and the PKA RIIβ surface was quantified using the Spot function of Imaris. Spots located within 1 μm or less from PKA RIIβ were classified as colocalizing with PKA.

## cAMP and PKA activity assay

For the cAMP assay, neurons were transduced with the Green Up cAMP biosensor (Montana Molecular) according to the manufacturer's instructions. On the day of imaging, neurons expressing cADDis/ExRai-AKAR2 biosensors alone or with HA-ACs were washed three times with pre-equilibrated HBS. 10 μM dopamine (Millipore Sigma) was added by bath application after a 1 min baseline, and 10 μM forskolin (Fsk, Millipore Sigma) and 500 μM 3-isobutyl-1-methylxanthine (IBMX, Millipore Sigma) were added 30 min later for 2 min. Images were taken every 20 s with a 488 nm laser. Image analysis was performed on unprocessed images using ImageJ Fiji[69]. Briefly, ROIs were drawn around individual cells, and the average fluorescence in the ROI was background subtracted and normalized to baseline. For cADDIs and non-targeted ExRai-AKAR2, fluorescence was then divided by the average fluorescence of the Fsk/IBMX treatment. Integrated cAMP and PKA responses were measured as the area under the curve of two phases (0–5 and 5–30 min of dopamine treatment) when indicated, or the phase 0-30 min, and normalized to the average HA-AC9 value. The peak value corresponds to the maximum fluorescence value between 0 and 30 min after dopamine addition and is normalized to the average HA-AC9 peak value. The plateau phase corresponds to the average fluorescence intensity between 20 and 30 min of dopamine treatment and is normalized to the average HA-AC9 phase value.

## Membrane AC assay

HEK293 cells were transfected with HA-AC9 or HA-AC9-LL > AA using Lipofectamine 2000 (Thermo Fisher). After 48 h, cells were rinsed, harvested, and lysed by Dounce homogenization in lysis buffer (20 mM HEPES, pH 7.4, 1 mM EDTA, 2 mM MgCl$_2$, 1 mM dithiothreitol, 250 mM sucrose) with protease inhibitors. Cells were subjected to centrifugation at $500 \times g$ to pellet nuclei, followed by centrifugation at $100{,}000 \times g$. Membranes were resuspended in lysis buffer and immediately used for protein determination and AC assays as described[58]. Membranes (15 μg) were incubated for 10 min at 30 °C in an AC assay mixture containing 5 mM Mg$^{2+}$, 500 μM ATP, and 100 μM of the PDE inhibitor Ro20-1724 in the absence or presence of 300 nM purified GTPγS-Gαs. Assays were stopped with an equal volume of 0.2 N HCl, and cAMP was detected by direct cAMP enzyme immunoassay using acetylation (Enzo Life Sciences).

## Statistical analysis and reproducibility

For each experiment, a biological replicate is defined as an independent neuronal culture, except for the cAMP and PKA activity assay where a biological replicate corresponds to an independent dish. All data are shown as mean ± standard error of the mean (s.e.m.) from at least three independent neuronal cultures, and images are representative of at least three independent neuronal cultures. Statistical analyses were performed using Prism 10 (GraphPad) using an unpaired *t*-test (or paired when indicated) to determine significance.

## Reporting summary

Further information on research design is available in the Nature Portfolio Reporting Summary linked to this article.

## Data availability

Source data are provided with this paper. The authors declare that the data supporting the findings of this study are available within the paper and its supplementary information files. Source data are provided with this paper.

## Code availability

Custom MATLAB codes used in this study have been deposited on GitHub and can be accessed at https://github.com/learipoll/Ripoll-et-al-2024 or at Zenodo [https://doi.org/10.5281/zenodo.13327675][71].

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

## Acknowledgements
We thank L. Lavis, A. Ehrlich, G. Peng and A. Marley for sharing reagents. We thank D. Jullié for assistance with primary neuron culture and for providing MATLAB scripts. We thank R.D. Mullins for the use of the iSIM microscope, and S. Lord and A. Charles-Orszag for advice and technical support. We thank members of the von Zastrow laboratory for valuable discussion and feedback on the manuscript. Imaging experiments were primarily carried out in the UCSF Center for Advanced Light Microscopy; we thank K. Herrington and S. Kim for their technical support and expertise. CSU-W1 Spinning Disk/High-Speed Widefield was supported by S10 Shared Instrumentation grant (1S10OD017993-01A1). We thank the Gladstone Histology and Light Microscopy Core, directed by B. Ndjamen, for assistance with image analyses. This study was supported by research grants from the National Institutes of Health (grant nos. DA012864, DA010711 and MH120212 to M.v.Z.; GM145291 to C.W.D). L.R. was supported by the European Molecular Biology Organization (ALTF 192-2019).

## Author contributions
L.R. and M.v.Z. conceived the project and designed experiments. L.R. carried out experiments and analyzed data. Y.L. and C.W.D. performed and analyzed the AC activity and expression experiments. L.R. and M.v.Z. and wrote the manuscript. All authors edited the manuscript.

## Competing interests
The authors declare no competing interests.
