## [Peer Review File · Nature Communications]

Spatial organization of adenylyl cyclase and its impact on dopamine signaling in neuronsREVIEWER COMMENTS

Reviewer #1 (Remarks to the Author):

Previously, the author has found that AC9 selectively targets to endosomes. This report submitted by Ripoll and Zastrow has identified a N-terminal molecular determinant that controls AC9 targeting to the endosome of primary cultured MSN. Although the authors have shown that AC9 endosomal localization may control PKA activity in the nucleus, the physiological function of AC9 and dopamine receptors in endosomes remains puzzling. The manuscript does not convincingly demonstrate how the findings could have a significant impact on the fields of cAMP signaling. In addition, the topic of this research is not very focused (AC3, AC5 and AC9).

While the research methodology is sound and the data are well-analyzed, the manuscript falls short in demonstrating strong impact that warrants publication in Nature Communication. However, with appropriate revisions, the work could find a more specialized journal for dissemination.

Reviewer #2 (Remarks to the Author):

While many studies have focused on the distribution of PDEs in the generation of localized cAMP signals, how the various isoforms of adenylyl cyclase contribute to these signals has been highly underappreciated. AC knockout animals display varied isoform-dependent phenotypes that can be explained by differences in compartmentalization of ACs, providing compelling rationale for the present investigation by Ripoll and von Zastrow. This paper is a further reminder of the general need for isoform-selective AC inhibitors and reliable antibodies, tools that have been sorely lacking in the field. A perhaps unavoidable limitation of the study, therefore, is its reliance on overexpression of the various ACs. Nevertheless, the authors provide many new and interesting insights in this work.

Using striatal neurons, the authors show that expressed AC3 localizes to primary cilia, AC5 localizes to cilia and the plasma membrane, while AC9 stably populates an internal vesicular compartment identified as endosomal. Most of the new information in the paper revolves around the function of this intracellular pool of AC9. The authors confirm that the dopamine receptor, D1R, localizes to cilia and plasma membrane, but also show that D1R slowly accumulates in the AC9-labeled vesicular compartment after dopamine stimulation. The AC5 N-terminus governs trafficking to the cilium while a di-leucine motif in the AC9 N-terminus is shown to be one of the determinants of endosomal accumulation of AC9. Mutation of this motif results in a partial reduction in endosomal AC9, with a shift to plasma membrane accumulation. These results are straightforward and convincing.

Some other results are less straightforward and could use additional explanation:

1. In figure 1, data for expressed AC isoform constructs are shown from at least n=3 independent cultures, but how many cilia does this represent?
2. Figure 5, overexpression of AC3 (and AC5) caused a general elevation of dopamine-stimulated cAMP and PKA (ExRai-AKAR2) relative to maximal stimulation with forskolin/IBMX. It's unclear whether AC3 was exclusively localized to cilia in these cells, or whether this reflects production from untargeted ACs?
3. It's interesting that AC9 overexpression did not affect global cAMP signals, at least not as measured by a commercial cAMP sensor, cADDIS, but did have a measurable effect on the PKA activity profile as measured by non-targeted ExRai-AKAR2. The use of NLS- and NES-ExRai-AKAR2 suggest that this difference is accounted for by a dominant effect of AC9 on PKA activity in the nuclear compartment. It's unfortunate that the authors are not using ExRai-AKAR2 in its intended ratiometric mode, as they are missing most of the sensitivity of the probe. Ratio images would be much more convincing and would provide information about any differences in baseline PKA activity.
4. The peak-plateau in PKA activity was different with WT AC9 vs. the mutant of AC9 lacking the dileucine motif that exhibited less targeting to endosomes. Based on prior work, the WT AC9 is presumably functional to produce cAMP at the endosomal-Golgi interface in striatal neurons upon dopamine addition. Is the AC9-LL>AA mutant competent to produce cAMP?
5. Some of the images in the paper don't suggest an especially high concentration of AC9-positive vesicles in the peri-nuclear region, or a Golgi-like distribution. The idea that PKA is getting preferentially activated in this zone and then PKA-cat shunted to the nucleus would be further supported by local measurements of cAMP and PKA using endosome-targeted biosensors.

Reviewer #3 (Remarks to the Author):

The authors present an intriguing study detailing the spatial localization of AC isoforms in striatal neurons. They confirm previous findings of AC3 and AC5 localization within cilia and present new findings on AC9 co-localization with the Dopamine 1 receptor (D1R) on endosomes. Little is known about the localization of adenylyl cyclase and its signaling from intracellular sites and thus this study represents an important and impactful advance. The authors define regions within the N-terminus of AC5 and AC9 that dictate their localization to cilia and endosomes, respectively. Mutations within this region results in loss of the respective localization to these subdomains but not the overall plasma membrane. This will be a key reagent for the field in the future. Importantly, reduction of endosomal AC9 by N-terminal mutation results in a reduced plateau phase of dopamine-induced cytosolic PKA sensor but not the cAMP sensor. The mutant AC9 also showed

lower nuclear PKA signaling. This is an interesting finding that will be highly impactful on the GCPR and cAMP fields. However, a few issues detract from what is overall a very good manuscript.

1. In the original AC9 trafficking paper by von Zastrow, AC9 was shown to traffic from the plasma membrane to endosomes in a Gs-dependent manner. However, the D1 dopamine receptor does not appear to drive any movement of AC9, rather AC9 appears as multiple independent pools. I applaud the use of primary cells for this study, but does AC9 not traffic in this system or is it that activation by the D1R is unique? What happens upon stimulation of b2AR or any of the GPCRs used in the prior study?

2. The authors show in Extended data fig1c that AC9 resides either on the endosomal membrane or appears on the endosome lumen. The latter is not consistent with the AC9 active site facing the cytosol for signaling as proposed, given that this is a C-terminal tag. Please explain.

3. Are images from Fig 2a, 2b treated with dopamine? They show significant endosomal localization, whereas, Fig 2c is largely PM localized prior to dopamine treatment. If not, perhaps some quantitation of endosomal localization for D1R with and without dopamine treatment is required.

4. Is the dileucine motif present in AC9 unique to that AC isoform?

5. It is not clear why global PKA but not cAMP would be altered by the AC9-LL>AA mutant. Does AC9-LL>AA mutation alter local endosomal-tagged cADD is measured cAMP activity?

5b. Does AC9-LL>AA mutation alter dopamine-stimulated CREB activation and/or transcriptional responses?

6. Please provide n number for Fig 6a-f and quantitation. The videos, although beautiful, lack any quantitation or negative controls. Does AC9-LL>AA alter AC9:PKA co-localization? Does AC5 lack any PKA co-localization at the Golgi?

7. Why does ExRai-AKAR2 in Fig 5g-i show clear differences in global cytosolic PKA signaling between AC9 and AC9-LL>AA while, ExRai-AKAR2-NES shows no difference. They should be measuring the same cytoplasmic PKA activity!

Minor:

1. The authors use two endosome markers, EEA1 and VPS35. Please elaborate on the types of endosomes that are differentially labeled for these markers, if any.
2. Please discuss the significance of AC3 and AC5 localization and subsequent effects on cAMP and PKA signaling as compared to AC9.

Reviewer #4 (Remarks to the Author):

This is an important study that has major significance for endosomal signaling and for the cAMP signaling community. The authors first address the targeting of the various adenylyl cyclase (AC) isoforms and focus not only on the organelle/plasma membrane localization but also explore different cell types. They focus on three C isoforms, AC3, AC5, and AC9, and convincingly show that only AC9 goes to endosomes and that recruitment to endosomes is agonist driven. This is extremely important as one completely misses the potential important physiological relevance of targeting when one looks only at 293 cells which is the typical model system for the community. Using a neuronal cell model (MSN) is thus a very significant feature of this study, and, in addition, the authors explore specifically the cAMP-mediated regulation that is driven by dopamine and the dopamine 1 receptor (D1R) receptor, including cAMP-mediated gene expression in the nucleus. In addition to showing specific localization to cilia vs. plasma membranes (AC3/AC5) and to ER and Golg (AC9), the authors identify the motifs that direct the targeting, They then go beyond this to look at the cAMP-regulation of dopamine signaling. So it is a rigorous study that has very important physiological relevance. So much cell-specific signaling is mediated by the various isoforms that sometimes differ not by their enzymatic signaling but by motifs that lie outside catalytic domains in regions that are typically intrinsically disordered. For PKA signaling this includes not only the ACs but also the PDEs, the regulatory and catalytic subunits of PKA and the PDEs. These are likely all assembled as highly specific cAMP signaling islands, and this paper adds significantly to our understanding of how such islands are assembled. Although there are some points that are glossed over which should really be addressed, overall this is a seminal contribution and publication is definitely recommended.

Specific comments to address:

1. While the authors do highlight the importance of the AC isoforms, they ignore one potentially important feature of the cAMP-signaling pathway that is especially important in neurons. They show that the RIIb subunit is most likely the regulatory subunit isoform that is involved in this pathway, they ignore the isoform classification of the catalytic C-subunit. The brain is unusual in the ~50% of the PKA signaling is mediated by Cb isoforms and we know very little about the specific pathways

that are driven by Cb. Signaling in the retina, for example, shows that Ca and Cb do not localize the same and that Cb may, in particular, play a role in organelles. While it is beyond the scope of this study to explore the Cb isoforms, which include a number of N-terminal splice variants, there are pan-specific antibodies for Ca and Cb. I think it would add a significant piece of information to this story if the authors could show if it is Ca or Cb that is associated with dopamine signaling in MSN cells.

2. In many cases the levels of these proteins can change significantly and, in some cases, relatively quickly in response to cAMP signaling. The levels of the RIIb subunit, in particular, can change significantly. Is there any evidence that the level of the RIIb isoform changes in response to the enhanced cAMP-mediated gene expression in the nucleus?

3. The authors ignore the AKAPs but there are a few such as AKAP10 (dual-specific AKAP2) and AKAP 11, and probably other, that are associated with vesicles. Are these candidates for participating in the assembly of the signaling complex in the endosome?

4. The authors show convincingly that the LL motif in the proximal part of the N-terminal region is important for localization to endosomes and show that there is also a signal buried in the distal region, residues 75-105. Have they screened any of the potential motifs that are buried in this distal region? Clearly there is some important information that is buried here, and this is completely new but left a bit dangling. If the authors look at the conservation of motifs in this region across species, do they see any motifs that are highly conserved? The dual WW motif is especially intriguing, and arginines also can play an important role. Isn't there some evidence that EE motifs are associated with endosomal signaling? Could the missing signal at least be more precisely localized with a set of deletion mutants or an AAAAA scan of this relatively small region. It would be interesting, at the very least, to include such an alignment of this region in different higher eukaryotes in the Supplement. Presumably targeting to endosomes is conserved in higher eukaryotes. Is dopamine signaling also conserved?

5. With respect to the N-terminal targeting region are there any disease mutations that could shed light on important motifs that are embedded in what is likely a highly disordered region? These intrinsically disordered regions are turning out to be extremely important for signaling. The C-terminal tails of the GPCRs, for example, are extremely important for signaling and these tails are typically simply deleted from the cryoEM structures.

Summary: Overall this is a very significant contribution, and publication if recommended. It would potentially add further significance to the manuscript if some of the above questions could be addressed.

Response to Referees (Ripoll et al, NCOMMS-23-54233-T)

We thank the reviewers for their thoughtful critiques and comments. We have addressed and responded to all of them, to the best of our ability, as summarized below in order that the comments were listed in the compiled review document. Substantial additions or changes in the text are indicated in the revised Article file in colored text.

Reviewer #1

Previously, the author has found that AC9 selectively targets to endosomes. This report submitted by Ripoll and Zastrow has identified a N-terminal molecular determinant that controls AC9 targeting to the endosome of primary cultured MSN. Although the authors have shown that AC9 endosomal localization may control PKA activity in the nucleus, the physiological function of AC9 and dopamine receptors in endosomes remains puzzling. The manuscript does not convincingly demonstrate how the findings could have a significant impact on the fields of cAMP signaling. In addition, the topic of this research is not very focused (AC3, AC5 and AC9).

While the research methodology is sound and the data are well-analyzed, the manuscript falls short in demonstrating strong impact that warrants publication in Nature Communication. However, with appropriate revisions, the work could find a more specialized journal for dissemination.

We appreciate that the reviewer found our methodology sound and data well-analyzed. We respectfully disagree with this reviewer's opinion regarding significance.

Reviewer #2

While many studies have focused on the distribution of PDEs in the generation of localized cAMP signals, how the various isoforms of adenylyl cyclase contribute to these signals has been highly underappreciated. AC knockout animals display varied isoform-dependent phenotypes that can be explained by differences in compartmentalization of ACs, providing compelling rationale for the present investigation by Ripoll and von Zastrow. This paper is a further reminder of the general need for isoform-selective AC inhibitors and reliable antibodies, tools that have been sorely lacking in the field. A perhaps unavoidable limitation of the study, therefore, is its reliance on overexpression of the various ACs. Nevertheless, the authors provide many new and interesting insights in this work.

Using striatal neurons, the authors show that expressed AC3 localizes to primary cilia, AC5 localizes to cilia and the plasma membrane, while AC9 stably populates an internal vesicular compartment identified as endosomal. Most of the new information in the

paper revolves around the function of this intracellular pool of AC9. The authors confirm that the dopamine receptor, D1R, localizes to cilia and plasma membrane, but also show that D1R slowly accumulates in the AC9-labeled vesicular compartment after dopamine stimulation. The AC5 N-terminus governs trafficking to the cilium while a dileucine motif in the AC9 N-terminus is shown to be one of the determinants of endosomal accumulation of AC9. Mutation of this motif results in a partial reduction in endosomal AC9, with a shift to plasma membrane accumulation. These results are straightforward and convincing.

Some other results are less straightforward and could use additional explanation:

We are pleased that the reviewer recognizes that our study addresses a highly under-appreciated aspect of cAMP signaling biology, and that they found our results to be generally straightforward and convincing. We are grateful for the insightful comments and critiques. We believe we have fully addressed these in the revised manuscript, as described below in line with each specific critique.

1. In figure 1, data for expressed AC isoform constructs are shown from at least n=3 independent cultures, but how many cilia does this represent?

This is a good point and we see that we had inadvertently not included this detail in the initial submission. For Fig 1 the numbers are:

Fig. 1d, n=3 biological replicates (independent cultures)
AC3=20 (replicate 1) +20 (replicate 2) +30 (replicate 3) = 70 cells total
AC5=23+20+30 = 73 cells
AC9=20+20+28 = 68 cells

Fig. 1e, n=3 (AC3, AC5) and n=4 (AC9)
AC3= 11+10+10+10 = 41 cells
AC5=12+10+10+11 = 43 cells
AC9=10+11+10 = 31 cells

We have included this information for all of the figures, added to each figure legend in the revised manuscript. We have added to the Methods section a description of how we define a biological replicate for each series of experiments (lines 555 - 557 in the revised manuscript) and have included details of statistical analysis in the source data file accompanying each figure.

2. Figure 5, overexpression of AC3 (and AC5) caused a general elevation of dopamine-stimulated cAMP and PKA (ExRai-AKAR2) relative to maximal stimulation with forskolin/IBMX. It's unclear whether AC3 was exclusively localized to cilia in these cells, or whether this reflects production from untargeted ACs?

While AC3 is highly concentrated in the cilium, it is also detectable elsewhere and we cannot presently exclude the presence of some protein in the extraciliary plasma

membrane. We agree that the question of ciliary vs extraciliary cAMP production is interesting but we have not addressed it in the present study. In the present study, we only measure total cAMP/PKA changes using untargeted biosensors and only compare AC3 and AC5 overexpression conditions to the untransfected condition, where cAMP production arises from endogenous ACs. Therefore, we cannot presently conclude whether the increase in cAMP and PKA activity is a result of cAMP production specifically in the cilium or from enzymes that are located in the extraciliary plasma membrane. Our focus here is on functional significance of AC9's targeting to endosomes, although we agree that the functional significance of AC3/5 targeting to cilia is also an interesting question for future study. We have added this important point to the Discussion section in the revised manuscript (lines 384 - 386).

3. It's interesting that AC9 overexpression did not affect global cAMP signals, at least not as measured by a commercial cAMP sensor, cADDIS, but did have a measurable effect on the PKA activity profile as measured by non-targeted ExRai-AKAR2. The use of NLS- and NES-ExRai-AKAR2 suggest that this difference is accounted for by a dominant effect of AC9 on PKA activity in the nuclear compartment. It's unfortunate that the authors are not using ExRai-AKAR2 in its intended ratiometric mode, as they are missing most of the sensitivity of the probe. Ratio images would be much more convincing and would provide information about any differences in baseline PKA activity.

We agree that ratiometric imaging can provide increased dynamic range of fluorescence detection, and that this is useful in many cases. For imaging neurons in the present study, however, we found single wavelength excitation sufficient to provide a robust readout (as supported by the present quantitative and statistical analysis). We have also found single wavelength imaging to speed acquisition with the imaging system presently available, enabling us to image a larger number of cells per experiment. It also reduces concerns about potential photobleaching or UV-induced phototoxicity of the neurons.

4. The peak-plateau in PKA activity was different with WT AC9 vs. the mutant of AC9 lacking the di-leucine motif that exhibited less targeting to endosomes. Based on prior work, the WT AC9 is presumably functional to produce cAMP at the endosomal-Golgi interface in striatal neurons upon dopamine addition. Is the AC9-LL>AA mutant competent to produce cAMP?

This is an excellent point. While we were confident from the previous data that the LL-mutant AC9 is functional, the reviewer is correct that we did not directly assess its enzymatic activity relative to the WT construct. We have added new data addressing this point to the revised manuscript in **Extended Data Fig 4**. Using a membrane AC activity assay, we demonstrate that the specific activities of WT AC9 and the LL-mutant AC9 (AC9-LL>AA) are indistinguishable, both at the level of basal and Gs-stimulated activity (**Extended Data Fig 4a**), and that the WT and LL-mutant constructs are expressed in neurons at a similar level (**Extended Data Fig 4b**).

5. Some of the images in the paper don't suggest an especially high concentration of AC9-positive vesicles in the peri-nuclear region, or a Golgi-like distribution. The idea

that PKA is getting preferentially activated in this zone and then PKA-cat shunted to the nucleus would be further supported by local measurements of cAMP and PKA using endosome-targeted biosensors.

These are both excellent points and we have addressed them in the revised manuscript as follows:

a) We agree that not all endosomes are in close proximity to PKA, and we do not claim this. However, we are confident that many endosomes present in the soma are in close proximity to PKA. In the revised manuscript, we have added new analysis demonstrating this fact (**Extended Data Fig 7**). Implementing an unbiased method of image segmentation and nearest-neighbor distance analysis (using “Spot” and “Surface” tools in Imaris), we verify that neurons contain a large number of endosomes (~30 on average) containing AC9 in the perinuclear region within 1 μm of PKA (RII β) accumulations (rendered as a surface by Imaris), and this is not true for AC5 (Extended Data Fig 7g). We have also clarified the important point in the revised Results section (lines 267 - 273).

b) We also agree that our model predicts that PKA activation occurs adjacent to AC9-containing endosomes. We have included new data in the revised manuscript attempting to test this (**Extended Data Fig 5**). Unfortunately, restrictions imposed by the company have prevented us from generating an endosome-targeted version of the cADDis cAMP sensor. However, we have generated an endosome-targeted version of ExRai-AKAR2 (ExRai-AKAR2-Endo) and verified robust localization. This validation is included in the revised manuscript in **Extended Data Fig 5b**. We compared the PKA activity measured by this sensor in neurons over-expressing WT relative to LL mutant AC9. We observed little or no difference in the initial (< 5 min) activity elevation, which we believe is due to the initial burst (before desensitization) of cAMP diffusing from the plasma membrane. However, WT AC9 consistently produced a higher activity signal at the endosome membrane than the LL mutant (reduced concentration in endosomes) at all later time points (> 5 min), a time window corresponding to the ‘plateau’ signaling phase when activated receptors accumulate in AC9-containing endosomes. This effect did not reach statistical significance using the same pooled analysis of peak and plateau phases used elsewhere in the study, so we only note it as a trend, but the trend is consistent at every individual later time point measured. These new data are included in the revised manuscript in **Extended Data Fig 5c**. We believe that these results provide additional support for the hypothesis of preferential activation around endosomes based on endosomal AC9 localization at later time points, while cAMP produced from the plasma membrane can activate nuclear PKA irrespective of AC9 at early time points. We acknowledge that our model, which is based only on local shunting of catalytic subunits into the nucleus by diffusion, is likely more complex (and does not explain the very rapid early component of activation from the plasma membrane). Nevertheless, as this model is sufficient to explain the later elevation of PKA activity in the nucleus that is sensitive to endosomal AC9 concentration, which is our main claim that we believe is very well-supported by the data, we have left it in place while adding clarification to the Results (lines 243 - 247 and 251 - 255).

Reviewer #3

The authors present an intriguing study detailing the spatial localization of AC isoforms in striatal neurons. They confirm previous findings of AC3 and AC5 localization within cilia and present new findings on AC9 co-localization with the Dopamine 1 receptor (D1R) on endosomes. Little is known about the localization of adenylyl cyclase and its signaling from intracellular sites and thus this study represents an important and impactful advance. The authors define regions within the N-terminus of AC5 and AC9 that dictate their localization to cilia and endosomes, respectively. Mutations within this region results in loss of the respective localization to these subdomains but not the overall plasma membrane. This will be a key reagent for the field in the future. Importantly, reduction of endosomal AC9 by N-terminal mutation results in a reduced plateau phase of dopamine-induced cytosolic PKA sensor but not the cAMP sensor. The mutant AC9 also showed lower nuclear PKA signaling. This is an interesting finding that will be highly impactful on the GCPR and cAMP fields. However, a few issues detract from what is overall a very good manuscript.

We are pleased that the reviewer found our manuscript to be interesting and of generally high quality. We thank the reviewer for the constructively critical comments, and we have done our best to address them as follows.

1. In the original AC9 trafficking paper by von Zastrow, AC9 was shown to traffic from the plasma membrane to endosomes in a Gs-dependent manner. However, the D1 dopamine receptor does not appear to drive any movement of AC9, rather AC9 appears as multiple independent pools. I applaud the use of primary cells for this study, but does AC9 not traffic in this system or is it that activation by the D1R is unique? What happens upon stimulation of β 2AR or any of the GPCRs used in the prior study?

This is an excellent point. AC9 localization to endosomes is indeed regulated in 293 cells but appears to be constitutive in neurons. As the reviewer also notes, our previous work on regulated AC9 localization was using the β 2AR, but not D1R. To address the question of whether the difference is cell type-specific or receptor-specific, we carried out additional experiments in striatal neurons testing the β 2AR in place of the D1R. We find AC9 still in endosomes irrespective of agonist-induced receptor activation, and that β 2ARs also accumulate in AC9-positive endosomes after activation. Therefore we are confident that the difference is cell type-specific rather than receptor-specific. We have included these new data in the revised manuscript as Extended Data Fig 2 e-f and note this important point in the revised Results. We have not yet identified a mechanistic basis for this difference and thus also point out, in the revised Discussion section (lines 335 - 338), this cell-specific difference in AC9 trafficking as an interesting future direction.

2. The authors show in Extended data fig1c that AC9 resides either on the endosomal membrane or appears on the endosome lumen. The latter is not consistent with the AC9

active site facing the cytosol for signaling as proposed, given that this is a C-terminal tag. Please explain.

We agree that our SIM images suggest that there is some AC9 in the endosome lumen, and that such localization is not consistent with a signaling function. We speculate that luminal AC9 represents a fraction that is enroute to lysosomes for degradation, but we are confident that there is also AC9 in the limiting membrane. To specifically test this, we carried out additional experiments labeling AC9 in its C-terminus with a pH-sensitive GFP variant whose fluorescence is quenched in the acidic environment of the endosome lumen (AC9-SEP). Using this tagging strategy, protein in the lumen has its fluorescence quenched and protein present in the limiting membrane (SEP exposed to the cytoplasm) is selectively detected. We previously validated this method for distinguishing GPCRs present in the limiting membrane from those present in the endosome lumen, as described and cited in the revised manuscript. These new data explicitly demonstrate the presence of AC9-SEP in the endosome limiting membrane and are included in Extended Data Fig 1d, and we have added corresponding explanations to the Methods (lines 445 - 446) and Results (lines 108 - 116) sections of the revised manuscript.

3. Are images from Fig 2a, 2b treated with dopamine? They show significant endosomal localization, whereas, Fig 2c is largely PM localized prior to dopamine treatment. If not, perhaps some quantitation of endosomal localization for D1R with and without dopamine treatment is required.

No, Fig. 2a and 2b are of cells not exposed to dopamine and we agree there is some endosomal receptor evident. D1R internalization is agonist-dependent, in that it is strongly stimulated by agonist, but there is a low level of constitutive internalization as well. However, this is not a large amount and the reason that it looks pronounced, relative to the other images in the figure, is that the images shown in Fig 2a are rendered from a maximum projection of a full z-stack whereas those shown in Fig 2c are rendered from a single section. Endosomes are thus over-represented because the maximum projection combines across sections. We are also confident that there is a significant dopamine-induced accumulation of D1R specifically in AC9-containing endosomes, as quantified in Fig 2d, and we verify this using EEA1 as a more standard endosome marker in Extended Data Fig 2b. We have clarified this difference between the rendering in the respective panels of Fig 2 in the legend of the revised manuscript.

4. Is the dileucine motif present in AC9 unique to that AC isoform?

AC3 does not have an N-terminal dileucine sequence but AC5 has one. However, in AC5, the dileucine is very close to the transmembrane domain while, in AC9, it is far from the transmembrane domain in sequence space. We show that AC5 does not concentrate in endosomes, whereas AC9 does, and that the AC9 N-terminus drives endosome localization when fused to replace the AC5 N-terminus. Therefore we are confident that the dileucine in AC9 is functional and speculate that its specific endocytic activity may be due to higher accessibility. There is precedent for such positional effects

of endocytic motifs (i.e. motifs located close to the membrane don't function) based on studies of other membrane proteins (eg PMID 9694892), but we can only speculate at present because (1) we have not specifically investigated the significance of distance from the membrane in ACs (2) the endocytic activity of dileucine motifs is characteristically sensitive to flanking residue context (PMID 12651740) which also differs between AC5 and AC9. The main conclusion that we claim is that the dileucine motif in AC9 is functional, and we believe this conclusion is well-supported by the present data.

5. It is not clear why global PKA but not cAMP would be altered by the AC9-LL>AA mutant. Does AC9-LL>AA mutation alter local endosomal-tagged cADDIs measured cAMP activity?

We think that AC9 present in endosomes makes a relatively small contribution to the total cellular cAMP production. We think it promotes PKA activation more efficiently due to location near perinuclear PKA stores. We think perinuclear PKA activation favors PKA activity accumulation in the nucleus also due to proximity. This is an admittedly simple model, and cannot rule out additional elements, but it is consistent with the present results and previous studies indicating an important role of location in determining the efficiency of PKA activation as cited in the text. In the revised manuscript (Results section), we have added additional clarification of this important point.

We agree that our model does predict a higher activity in the immediate vicinity of AC9-containing endosomes. As noted above in response to reviewer #2 (response 5b), endosomal cADDIs is not available to us presently, but we have generated an endosomal version of the PKA sensor (ExRai-AKAR2-Endo) and carried out additional experiments using it (Extended Data Fig 5). We think that the results are consistent with our model (a trend toward higher local PKA activity with AC9 concentration in endosomes) although, as noted also above, we acknowledge that there could be additional aspects to the specificity of signaling from endosomes that we have yet to delineate.

5b. Does AC9-LL>AA mutation alter dopamine-stimulated CREB activation and/or transcriptional responses?

This is an interesting question but we have not investigated it. In the present study we have restricted our focus to establishing isoform-selective targeting of AC in neurons and elucidating effects of endosomal AC9 localization on PKA activity. We agree that the results shown are consistent with effects also on CREB activation or CREB-dependent transcription. We have not carefully tested this yet and intend to pursue this in the future, as part of a larger survey of location - dependent downstream signaling effects.

6. Please provide n number for Fig 6a-f and quantitation. The videos, although beautiful, lack any quantitation or negative controls. Does AC9-LL>AA alter AC9:PKA co-localization? Does AC5 lack any PKA co-localization at the Golgi?

We are pleased that the reviewer appreciates the overall image quality, and the criticism regarding lack of quantification is fair. We show a single surface rendering in Fig 6 a-d, as the reviewer notes. We are confident that it is representative and, to demonstrate this, we have included new quantitative analysis of the colocalization of AC relative to PKA. Using an unbiased approach, we have measured the number of AC-positive endosomes in close proximity (within 1 μm) to the internal PKA compartment from three independent biological replicates (Extended Data Fig 7). We have also added description of these new data in the revised Methods (lines 531 - 535) and Results (lines 267 - 274) sections. We observed a pronounced degree of colocalization for both AC9 WT and the LL mutant. There is a trend toward a decrease in colocalization of the LL mutant, but the LL mutant still localizes in proximity with PKA. AC5, in contrast, does not. We believe that these results are fully consistent with our other data indicating that mutation of the dileucine motif in AC9 decreases its concentration in endosomes but does not fully prevent the AC targeting to endosomes. Further, as we see almost no colocalization of AC5 with PKA using the same analysis, we believe this further verifies the reported isoform-selectivity of AC targeting.

7. Why does ExRai-AKAR2 in Fig 5g-i show clear differences in global cytosolic PKA signaling between AC9 and AC9-LL>AA while, ExRai-AKAR2-NES shows no difference. They should be measuring the same cytoplasmic PKA activity!

This is a good point that we now see was not adequately addressed. The reason for the difference noted by the reviewer is that ExRai-AKAR2 localizes to the cytoplasm but it is also present in the nucleus in sufficient amount to detect nuclear activity elicited by endogenous D1R activation. Accordingly, we think that the untargeted ExRai-AKAR2 biosensor effectively detects combined nuclear and cytoplasmic activity. This is why we were able to use ExRai-AKAR2 to quantify nuclear activity in Extended Data Fig 9. For this reason, we generated the additional NES construct that is highly excluded from the nucleus and detects activity in the cytoplasm selectively. The overall conclusion that we draw from these results is that PKA activity in the cytoplasm is insensitive to AC9 localization in endosomes, whereas activity in the nucleus is specifically sensitive. We have added clarification about this important point in the revised Results section (lines 285 - 287) and included a new supplementary movie (**Supplementary Video 4**) showing a representative field in which that ExRai-AKAR2 fluorescence in several neurons is seen first in the cytoplasm and then in the nucleus.

Minor:

1. The authors use two endosome markers, EEA1 and VPS35. Please elaborate on the types of endosomes that are differentially labeled for these markers, if any.

We use EEA1 as a widely accepted marker of the early endosome compartment. In our hands, in striatal neurons VPS35 labels a similar compartment because most VPS35-marked puncta also are positive for EEA1. VPS35, as part of the retromer complex, labels a distinct subdomain of the endosome limiting membrane. It is also thought to label a late endocytic compartment, also in a subdomain, but we use VPS35 and EEA1 almost interchangeably in neurons because their localization patterns largely overlap.

We have observed this before in studies of GPCR trafficking in neurons, and we wanted to use both markers in the present study to provide additional validation of AC9 localization to a relevant endosomal compartment. In the revised manuscript we clarify that both are established endosome markers in this cell type (lines 100 - 101).

2. Please discuss the significance of AC3 and AC5 localization and subsequent effects on cAMP and PKA signaling as compared to AC9.

We have investigated the functional significance of AC localization in the present study only for endosomes and AC9. We have not assessed the functional significance of AC3 and AC5 localization between the cilium and extraciliary plasma membrane. We are interested to do so but so far have been unable to identify appropriate mutants to cleanly manipulate localization without other effects. The N-terminal truncation of AC5 (AC5- Δ Nter, see Fig.3c) is useful for trafficking studies but we are reluctant to interpret signaling data with it because the AC5 N-terminus makes a functionally significant interaction with G protein $\beta\gamma$ subcomplexes (pmid 26206488, 38589608).

Reviewer #4 (Remarks to the Author):

This is an important study that has major significance for endosomal signaling and for the cAMP signaling community. The authors first address the targeting of the various adenylyl cyclase (AC) isoforms and focus not only on the organelle/plasma membrane localization but also explore different cell types. They focus on three C isoforms, AC3, AC5, and AC9, and convincingly show that only AC9 goes to endosomes and that recruitment to endosomes is agonist driven. This is extremely important as one completely misses the potential important physiological relevance of targeting when one looks only at 293 cells which is the typical model system for the community. Using a neuronal cell model (MSN) is thus a very significant feature of this study, and, in addition, the authors explore specifically the cAMP-mediated regulation that is driven by dopamine and the dopamine 1 receptor (D1R) receptor, including cAMP-mediated gene expression in the nucleus. In addition to showing specific localization to cilia vs. plasma membranes (AC3/AC5) and to ER and Golg (AC9), the authors identify the motifs that direct the targeting. They then go beyond this to look at the cAMP-regulation of dopamine signaling. So it is a rigorous study that has very important physiological relevance. So much cell-specific signaling is mediated by the various isoforms that sometimes differ not by their enzymatic signaling but by motifs that lie outside catalytic domains in regions that are typically intrinsically disordered. For PKA signaling this includes not only the ACs but also the PDEs, the regulatory and catalytic subunits of PKA and the PDEs. These are likely all assembled as highly specific cAMP signaling islands, and this paper adds significantly to our understanding of how such islands are assembled. Although there are some points that are glossed over which should really be addressed, overall this is a seminal contribution and publication is definitely recommended.

We are pleased that this reviewer found our work important and interesting. We thank the reviewer for insightful comments and critiques. We have done our best to address them, as explained below, and also to point out important directions for future study.

Specific comments to address:

1. While the authors do highlight the importance of the AC isoforms, they ignore one potentially important feature of the cAMP-signaling pathway that is especially important in neurons. They show that the RIIb subunit is most likely the regulatory subunit isoform that is involved in this pathway, they ignore the isoform classification of the catalytic C-subunit. The brain is unusual in the ~50% of the PKA signaling is mediated by Cb isoforms and we know very little about the specific pathways that are driven by Cb. Signaling in the retina, for example, shows that Ca and Cb do not localize the same and that Cb may, in particular, play a role in organelles. While it is beyond the scope of this study to explore the Cb isoforms, which include a number of N-terminal splice variants, there are pan-specific antibodies for Ca and Cb. I think it would add a significant piece of information to this story if the authors could show if it is Ca or Cb that is associated with dopamine signaling in MSN cells.

This is a great point, and we agree that PKA subtype diversity is likely a very important issue in the brain. To answer the reviewer's question, the localization of the endogenous catalytic subunit that we show (in Fig 6a and Extended Data Fig 6) is of C α because the antibody used is specific for this isoform and has been used previously to detect endogenous C α in striatal tissue (PMID 32357495). In the live cell images we have used recombinant C α for consistency. While we are confident that C α is endogenously expressed in these neurons and localizes with RII β at the Golgi apparatus, we cannot exclude an additional or distinct role of C β . We therefore believe that C α marks the endosome-adjacent PKA pool, which is a key point relevant to our conclusions, but we cannot say whether this is true also for C β . We clarify the identity of the catalytic subunit localized in the revised Results section (line 262) and point to PKA subtype diversity, with additional citation as suggested, in the Discussion (lines 359 - 360) as an important future direction.

2. In many cases the levels of these proteins can change significantly and, in some cases, relatively quickly in response to cAMP signaling. The levels of the RIIb subunit, in particular, can change significantly. Is there any evidence that the level of the RIIb isoform changes in response to the enhanced cAMP-mediated gene expression in the nucleus?

We were not aware of this information and agree that modulation of PKA subunit abundance may be another important aspect. We did not observe any clear evidence for changes in endogenous RII β staining by fluorescence microscopy but have not quantified the staining intensity, as would be required to address this in the present study.

3. The authors ignore the AKAPs but there are a few such as AKAP10 (dual-specific AKAP2) and AKAP 11, and probably other, that are associated with vesicles. Are these candidates for participating in the assembly of the signaling complex in the endosome?

We don't mean to ignore AKAPs (!), and acknowledged in the Introduction that AKAPs are key players in localizing PKA, ACs and PDEs. We don't know if a specific AKAP contributes to endosome localization of AC9, but we doubt it because we do not detect PKA localization to endosomes in striatal neurons (PKA localizes to Golgi membranes but AC9 is on endosomes). Certainly AKAPs are known to localize PKA at the Golgi apparatus and, in the present study, we verify PKA-RII β and C α localization to Golgi membranes. However, our focus is on the fact that AC9 is on a distinct, but closely adjacent, membrane compartment and on the functional role of endosomal AC9 in promoting PKA signaling to the nucleus. We agree that further defining AKAP(s) that organize PKA adjacent to AC9-containing endosomes, likely through interactions with membranes of the Golgi apparatus, is an interesting direction for future study.

4. The authors show convincingly that the LL motif in the proximal part of the N-terminal region is important for localization to endosomes and show that there is also a signal buried in the distal region, residues 75-105. Have they screened any of the potential motifs that are buried in this distal region? Clearly there is some important information that is buried here, and this is completely new but left a bit dangling. If the authors look at the conservation of motifs in this region across species, do they see any motifs that are highly conserved? The dual WW motif is especially intriguing, and arginines also can play an important role. Isn't there some evidence that EE motifs are associated with endosomal signaling? Could the missing signal at least be more precisely localized with a set of deletion mutants or an AAAAA scan of this relatively small region. It would be interesting, at the very least, to include such an alignment of this region in different higher eukaryotes in the Supplement. Presumably targeting to endosomes is conserved in higher eukaryotes. Is dopamine signaling also conserved?

The reviewer is correct that the dileucine motif is not the only endocytic determinant in the AC9 N-terminus, and the results shown in the initial submission identified endocytic activity in the distal third of the N-terminus as well as in the proximal third that contains the dileucine motif. To address the reviewer's question, we carried out additional mutational studies to define the determinant in the distal region. We have identified a second endocytic determinant, distinct from the dileucine and that does not correspond to any previous identified trafficking determinant (LEEACL). Mutation of this sequence also causes ER retention, however, so we have avoided mutations in this region for functional studies. These new data are presented in **Extended Data Fig 3** of the revised manuscript and described in an additional paragraph in the Results section (lines 201 - 220).

5. With respect to the N-terminal targeting region are there any disease mutations that could shed light on important motifs that are embedded in what is likely a highly disordered region? These intrinsically disordered regions are turning out to be extremely important for signaling. The C-terminal tails of the GPCRs, for example, are extremely

important for signaling and these tails are typically simply deleted from the cryoEM structures.

We are not aware of disease-associated mutations or polymorphisms affecting the N-terminal sequence, but there are polymorphisms that affect other regions of AC9. We agree with the reviewer that intrinsically disordered regions are turning out to be very important, despite the challenges in studying them.

Summary: Overall this is a very significant contribution, and publication is recommended. It would potentially add further significance to the manuscript if some of the above questions could be addressed.

We are pleased that the reviewer appreciates the value of our study, and thank the reviewer for insightful comments and advice for increasing its reach.

REVIEWER COMMENTS

Reviewer #2 (Remarks to the Author):

The authors have done a great job with the revisions. I have no further comments on the content of the paper. In the future these investigators might want to check out the wealth of excellent genetically-encoded cAMP sensors that are now available to circumvent the problem of not being able to modify the commercial cADDIS reporter. New data with targeted cAMP reporters would have been the ultimate addition to an already excellent report.

Reviewer #4 (Remarks to the Author):

The authors have seriously and satisfactorily addressed most of the concerns raised by the reviewers. Three of the reviews were extensive and raised valid points. I am satisfied that the responses are adequate and consider that the manuscript is now suitable for publication.

Reviewer #5 (Remarks to the Author):

The manuscript „Spatial organization of adenylyl cyclase and its impact on dopamine signaling in neurons” by Mark von Zastrow’s lab describes the subcellular distribution and functional consequences of membrane-bound adenylyl cyclases, the critical enzymes that produce most of the cellular cAMP. cAMP is a pivotal second messenger that relays the information of hundreds of GPCRs throughout the cell. It has been well established that cAMP signaling is highly compartmentalized in intact cells; and this compartmentalization is essential for cells to elicit many functions simultaneously with the same second messenger. As very little is known about the role of ACs in cAMP compartmentalization, it is unquestionable that this manuscript is of great importance and provides a conceptual advance for the field.

I have not seen the original version of the manuscript but in the present revised version the data appear sound and well interpreted. All images from various microscopy techniques look beautiful and provide a visually accessible way to understand the key messages of the manuscript. I believe that this manuscript is very appropriate for Nature communications.

With the intention to further strengthen the manuscript and increase its impact, I list some suggestions:

- 1) The manuscript would benefit from using a ratiometric cAMP biosensor that can be targeted to endosomes, to the Golgi and/or to the perinuclear region. This would allow tracking cAMP that is produced at endosomes in real-time. I appreciate the use of endo-ExRaiAKAR, however, monitoring cAMP itself would be more direct.
- 2) The concept of membrane contact sites is very intriguing. In my understanding the authors suggest that cAMP is produced at endosomes which is then somewhat 'channeled' to PKA residing in the perinuclear region. It would be informative to record PKA activity at the perinuclear region to study the dynamics and kinetics of cAMP signaling at such membrane contact sites. How is this process tuned? Is there an AKAP involved that links endosomes to the perinuclear region?
- 3) Given the pivotal role of endosomal AC9 to regulate nuclear signaling, I was wondering whether AC9 knock-out mice show defects in learning processes?

On a more technical note: some data in Figure 5 and Extended Data Figure 5 do not look normally distributed or cluster in two distinct populations (e.g. Fig 5b, e, f). I fear this may confound the interpretation of the data.

Minor: The authors argue in their introduction that cAMP is compartmentalized in nanodomains and that compartmentalized cAMP signaling requires local cAMP production despite the diffusible nature of cAMP. We have shown previously that cAMP does not diffuse freely at physiological concentrations and that this allows nanodomain formation by PDEs. It would be great if this work could be acknowledged (PMID: 32846156).

Andreas Bock
Leipzig University
Germany

Reviewer #6 (Remarks to the Author):

The manuscript authored by Ripoll et al has undergone a thorough revision by the authors, who have demonstrated a commendable commitment to addressing the main concerns raised during the review process. Notably, they have incorporated new data, significantly enhancing the quality of the manuscript and reinforcing its underlying hypothesis. Furthermore, the authors have provided comprehensive clarifications regarding the methodology, thereby ensuring transparency and reproducibility in their research approach.

The findings presented in the manuscript hold substantial importance within the field of cAMP and GPCR, offering valuable insights that can greatly impact the field. Given the rigor of the revisions and the significance of the research outcomes, I strongly endorse the publication of this manuscript.

RESPONSE TO REVIEWERS

We are pleased that all of the reviewers found our manuscript improved and raised no additional critiques. We therefore limit the present response to suggestions and critiques from Reviewer #5, who had not seen the manuscript previously and raised additional points.

Reviewer #5 (Remarks to the Author):

The manuscript „Spatial organization of adenylyl cyclase and its impact on dopamine signaling in neurons” by Mark von Zastrow’s lab describes the subcellular distribution and functional consequences of membrane-bound adenylyl cyclases, the critical enzymes that produce most of the cellular cAMP. cAMP is a pivotal second messenger that relays the information of hundreds of GPCRs throughout the cell. It has been well established that cAMP signaling is highly compartmentalized in intact cells; and this compartmentalization is essential for cells to elicit many functions simultaneously with the same second messenger. As very little is known about the role of ACs in cAMP compartmentalization, it is unquestionable that this manuscript is of great importance and provides a conceptual advance for the field.

We are pleased that Reviewer #5 recognizes the overall importance of determining how AC is spatially distributed in cells, and that our study makes a significant advance in this direction.

I have not seen the original version of the manuscript but in the present revised version the data appear sound and well interpreted. All images from various microscopy techniques look beautiful and provide a visually accessible way to understand the key messages of the manuscript. I believe that this manuscript is very appropriate for Nature communications.

With the intention to further strengthen the manuscript and increase its impact, I list some suggestions:

We have done our best to address the Reviewers’ insightful suggestions, within the constraints of what we are able to achieve using presently available resources, as follows:

1) The manuscript would benefit from using a ratiometric cAMP biosensor that can be targeted to endosomes, to the Golgi and/or to the perinuclear region. This would allow tracking cAMP that is produced at endosomes in real-time. I appreciate the use of endo-ExRaiAKAR, however, monitoring cAMP itself would be more direct.

We agree that local cAMP imaging using a ratiometric sensor would provide more direct information, and recognize that Prof. Bock’s group has pioneered methods for measuring such local concentrations with high spatial and temporal resolution. We intend to extend our experimental approach in this direction in future experiments but, at present, our imaging setup requires sequential image acquisitions in order to determine local fluorescence ratios. We have found this incompatible with accurate ratio determination due to movement artifacts that occur across sequential frames. We are aware that this issue could be addressed with truly

simultaneous imaging of fluorescence channels and / or faster image acquisition, but we have not yet successfully implemented such methodological improvements in our lab.

In the present study, we have focused on establishing that ACs are differentially localized in an isoform-specific manner, that AC9 is selectively concentrated in endosomes, and that endosomal concentration promotes the ability of endogenous D1R activation to elevate PKA activity in the nucleus. We think that the intensity-based measurement of PKA activity that we have used is sufficient to identify the latter functional effect, but we definitely agree that ratiometric cAMP imaging would be more direct and an essential approach for moving forward beyond the present conclusions. In the revised manuscript (lines 359-362), we explicitly note this as a future direction, and cite Bock et al as an elegant example of how such measurements can be achieved with nanometer-scale spatial resolution.

2) The concept of membrane contact sites is very intriguing. In my understanding the authors suggest that cAMP is produced at endosomes which is then somewhat 'channeled' to PKA residing in the perinuclear region. It would be informative to record PKA activity at the perinuclear region to study the dynamics and kinetics of cAMP signaling at such membrane contact sites. How is this process tuned? Is there an AKAP involved that links endosomes to the perinuclear region?

We are also intrigued by how such local membrane apposition is achieved and how the signaling is tuned.

To the first point, some membrane contact sites are formed by bridging protein interactions that physically connect adjacent membranes in trans. We are not aware of any such bridges in the present example, and note that not all membrane contacts have such a known biochemical basis (e.g., see PMID 31732717). Nevertheless, there is clear precedent for endosome membranes being in close apposition to Golgi or TGN membrane elements (e.g. see PMID 31974113, where a subset of recycling endosomes were shown to contact TGN). Again, there is no known established biochemical bridge between the membranes, and a simple possibility is that proximity is achieved by microtubule-based motility driving the endosomes toward the centrosome / MTOC around which the Golgi apparatus is organized. It is also conceivable that proximity is mediated by an AKAP, as the Reviewer suggests. We are aware of one AKAP protein, Yotiao, that binds AC9 directly. However, we are not aware of any evidence that this protein is capable of mediating cross-membrane bridges. In the revised manuscript, we do not speculate on any particular mechanism, and we propose this question as an important direction for future investigation (lines 375-376).

To the second question, how the signal is tuned, we agree this is also intriguing. Briefly summarized, we strongly believe that cAMP production from endosomes is tightly regulated (and in a GPCR-specific manner) but we don't really understand how. Previous work from Jeff Benovic's lab and ours identified the residence time of receptors in endosomes before they recycle as a key variable (PMID 27226565), but how this is achieved remains poorly understood and control of receptor residence time in endosomes is unlikely to be the full explanation. In the

present study, we show that AC9-containing endosomes are in close proximity to PKA concentrated on Golgi membranes irrespective of D1R activation, but we have not tested if D1R activation has an additional effect. We do not speculate on these ideas in the present study, but agree that this question is important for future study.

3) Given the pivotal role of endosomal AC9 to regulate nuclear signaling, I was wondering whether AC9 knock-out mice show defects in learning processes?

This is another fascinating question. As far as we know, nobody has looked specifically for learning deficits in AC9-deficient mice. An AC9 KO mouse exists and was characterized by the Dessauer lab in 2017 with a focus on cardiac defects (PMID 28717248). Here, it was noted that the KO animals have reduced viability and Ferenc Antony's group also noted reduced KO viability. Therefore we think a tissue-specific or conditional KO strategy would be needed. To our knowledge, this has not yet been attempted or achieved for AC9. We do agree that this is an interesting question in light of the present results with MSNs, in which AC9 is quite prominently expressed but has no presently identified functional significance.

On a more technical note: some data in Figure 5 and Extended Data Figure 5 do not look normally distributed or cluster in two distinct populations (e.g. Fig 5b, e, f). I fear this may confound the interpretation of the data.

We agree that the data spread is pretty wide in some of the signaling experiments using neurons, and we think that there are multiple sources of variability that contribute to this. To the Reviewer's specific comment, we have tested for normality of the data distributions presented in Fig 5 and in Ext Data Fig 5. The data in Fig 5b, for example, pass normality testing using D'Agonisto and Pearson. We are also not surprised to see the elevation of cAMP and PKA activity reported, as AC5 was previously shown to be a main contributor to cellular cAMP production in MSNs. We could not apply D'Agonisto and Pearson to Ext Data Fig 5 because the statistical N is not sufficiently large, but data in that figure pass the Shapiro-Wilk test. We are not experts in statistics, however, and we agree that we might be missing something in the variability. Nevertheless, as we have observed an effect of AC9 endosomal concentration on PKA activity both using an untargeted sensor (Fig 5) and using targeted versions in independent experiments (Fig 6), we are confident that there is indeed a functional effect at the level of the later-phase 'plateau' signal impacting nuclear PKA activity.

Minor: The authors argue in their introduction that cAMP is compartmentalized in nanodomains and that compartmentalized cAMP signaling requires local cAMP production despite the diffusible nature of cAMP. We have shown previously that cAMP does not diffuse freely at physiological concentrations and that this allows nanodomain formation by PDEs. It would be great if this work could be acknowledged (PMID: 32846156).

This is a good point that we did not mean to overlook, but we mentioned local cAMP diffusion gradients in a very general way, and with citation of a single review. In the revised manuscript, we have expanded that introductory sentence and included the citation noted (lines 36-38).